# GAIR: A Multimodal Geo-Foundation Model with Geo-Aligned Implicit Representations

## Abstract

**Vision Transformer (ViT)** has been used in many computer vision tasks with excellent results by providing representations for a whole image or image patches. However, ViT lacks *detailed localized image representations* at arbitrary positions when applied to geospatial tasks that involve multiple geospatial data modalities, such as overhead remote sensing (RS) data, ground-level imagery, and geospatial vector data. Here high-resolution localized representations are vital for modeling geospatial relationships and alignments across modalities. We proposed to solve this representation problem with an implicit neural representation (**INR**) module extending ViT with Neural Implicit Local Interpolation, which produces a continuous RS image representation covering arbitrary location in the RS image. Based on the INR module, we propose **GAIR**, a multimodal Geo-Foundation Model (GeoFM) integrating overhead RS data, street view (SV) imagery, and their geolocation metadata. GAIR utilizes three factorized neural encoders to project different modalities into the embedding space, and the INR module is used to further align these representations geographically, which are trained with contrastive learning objectives from unlabeled data. We evaluate GAIR across 9 geospatial tasks and 22 datasets spanning RS image-based, SV image-based, and location embedding-based benchmarks. Experimental results demonstrate that GAIR outperforms state-of-the-art geo-foundation models and alternative training objectives (e.g., MoCo-V2 and MAE) that do not use fine-grained geo-aligned spatial representations. Our results highlight the effectiveness of GAIR in **learning generalizable geospatial representations across tasks, spatial scales, and temporal contexts.**

## 1 Introduction

In the geospatial domain, there is a vast amount of **unlabeled geospatial datasets** such as satellite images, street view images, and user-generated geo-tagged data (e.g., Flickr images, geo-tagged tweets, iNaturelist species images, etc). In contrast, **labeled geospatial data** is typically scarce and highly imbalanced in terms of spatial, temporal, and class coverage (Mai et al., 2023a; Klemmer et al., 2023) due to the high cost of data annotation and the specialized domain expertise required. This scarcity of labeled data significantly limits the usage of these multimodal geospatial data in critical geospatial applications such as economic development prediction (Jean et al., 2016), species distribution

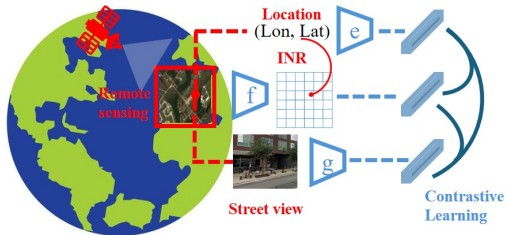

Figure 1: Overview of the **GAIR** architecture. The model encodes three modalities: street view image $s_i$, geolocation $x_i$, and remote sensing image $r_i$. ViT-based encoders extract features $g(s_i)$ and $f(r_i)$, while an implicit location encoder maps $x_i$ to $e(x_i)$. An INR module refines $f(r_i)$ into a geo-aligned embedding $z_i^q$, which is used for contrastive learning with $g(s_i)$ and $e(x_i)$.

modeling (Mai et al., 2023b; Cole et al., 2023), crop yield estimation (Azzari et al., 2017; You et al., 2017), urban dynamics monitoring (Cai et al., 2020), geographic question answering (Mai et al., 2020a; Yu et al., 2025), and climate extreme event detection (Ham et al., 2019). Furthermore, training AI models on these limited labeled datasets constrains their generalizability across space (Li et al., 2022a; 2023a; Wu et al., 2024), time, and task (Cong et al., 2022; Mai et al., 2025). Meanwhile, inspired by the recent advancements of language and vision foundation models, geo-foundation models

(GeoFMs) (Mai et al., 2024; Hsu et al., 2024; Janowicz et al., 2025) are developed as task-agnostic pre-trained models to tackle the issue of limited supervision information. However, *existing GeoFMs heavily rely on overhead remote sensing (RS) tasks* (Cong et al., 2022; Guo et al., 2024; Hong et al., 2024), e.g., RS foundation models (RSFMs). Although multiple multimodal GeoFMs (Fuller et al., 2024; Astruc et al., 2024) have been developed, they primarily focus on integrating different modalities of overhead RS data (e.g., optical, synthetic aperture radar (SAR), etc.) and/or language modality (Kuckreja et al., 2024; Zhang et al., 2024) while overlooking other important geospatial data modalities such as ground-level images and geospatial vector data. This practice significantly limits the spatial reasoning capabilities of these models and hampers their generalizability across different tasks, regions, spatial scales, and temporal contexts. Furthermore, existing GeoFMs mainly rely on conventional language and vision foundation model objectives like temporal or spatial augmented embeddings (Ayush et al., 2021; Stojnic & Risojevic, 2021; Guo et al., 2024) or image reconstruction (Cong et al., 2022; Sun et al., 2022; Jain et al., 2022), without explicitly considering the geospatial relationships across different data modalities, leading to suboptimal results for geospatial applications.

A key challenge for the current RS-based GeoFMs is aligning geospatial data across diverse spatial scales. For instance, overhead satellite imagery provides a broad spatial contextual view, while street view (SV) images capture fine-grained ground-level details. A single $256 \times 256$ overhead Sentinel-2 RS image can correspond to hundreds or even thousands of SV images at different geolocations, which makes it extremely difficult to create valuable geo-aligned RS-SV image pairs for the self-supervised learning (SSL) purpose. Traditional SSL paradigms (He et al., 2020; 2022) struggle to effectively align these disparate scales effectively, hindering the integration of multimodal geospatial data. Moreover, these SV images may not be located at the exact RS image pixel/patch centers while commonly used Vision Transformer (ViT) encoders (Dosovitskiy, 2020) only produce representations for a whole image or image patches, lacking *detailed localized image representations* needed to align geospatial data across diverse spatial scales.

In this paper, inspired by resolution-agnostic image super-resolution methods such as LIIF (Chen et al., 2021b), we proposed to leverage neural implicit functions(Sitzmann et al., 2020; Chen et al., 2021b; Gao et al., 2023) to learn a continuous image representation of an RS image and extract a localized neural representation that is geographically colocated with a ground-level image. By further projecting the SV image and its geolocation into an SV image embedding and a location embedding (Mai et al., 2020c; 2023a; Klemmer et al., 2023; Wu et al., 2024) with respective neural encoders, we can form a self-supervised learning objective by performing multi-objects contrastive learning on these three geo-aligned embeddings. Based on this geo-aligned contrastive learning framework, we develop a multimodal geo-foundation model called GAIR as shown in Figure 1.

The contributions of this paper are as follows:

- We propose GAIR, a GeoFM capable of handling diverse geospatial modalities, including remote sensing imagery, street view imagery, and geo-locations. By integrating these modalities, GAIR can perform a more comprehensive geospatial understanding and reasoning.

- The design of GAIR involves three key technical components: a) Factorized multimodal encoder design allows the model to retain modality-specific information while effectively learning cross-modal relationships. b) Neural implicit functions learn continuous representations on RS images and extract fine-grained localized embeddings that are geographically co-located with ground-level images. c) Contrastive learning is utilized on these geo-aligned neural embeddings from three distinct modalities in order to learn generalizable geospatial representations across different tasks.

- We pre-train GAIR on a globally sampled dataset named **Streetscapes1M** with 1 million sampled tuples. The pre-trained GAIR is adapted to a wide range of geospatial tasks via few-shot learning or fully fine-tuning. Experimental results show that GAIR outperforms all baselines and achieves state-of-the-art (SOTA) performance across all 9 geospatial tasks and 22 datasets, including street view imagery tasks, remote sensing imagery tasks, and location embedding tasks.

- Further analysis shows that geo-alignment SSL is crucial to achieving superior performance on single-modal tasks, and multimodal fusion can further add 4-20% performance gain. While Streetscapes1M is biased towards urban areas, by further pretraining GAIR on an urban-rural balanced dataset, the geographic bias of GAIR can be significantly reduced without sacrificing performance. Further qualitative and quantitative analysis show that GAIR can capture spatial relations across modalities, which is critical for diverse tasks such as image geolocalization.

## 2 RELATED WORK

**Geo-Foundation Models.** Many recent Geo-Foundation Models (GeoFMs) draw inspiration from Vision Foundation Models (Deng et al., 2009; Chen et al., 2020; Caron et al., 2021; Grill et al., 2020; Liu et al., 2021; Chen et al., 2021a; He et al., 2022; Oquab et al., 2023) and Vision-Language Models (Radford et al., 2021; Zhang et al., 2021; Wang et al., 2023). Unlike traditional vision datasets, geospatial data inherently integrates spatial and temporal information, requiring specialized model architectures. Current GeoFMs can be roughly categorized into (1) Remote Sensing Foundation Models (RSFMs), (2) Weather and Climate Foundation Models, and (3) geospatial vision-language foundation models (GeoVLFMs). RSFMs leverage contrastive learning or masked image modeling (MIM) to learn task-agnostic neural representations. GASSL (Ayush et al., 2021) and SeCo (Manas et al., 2021) use temporal augmentations with a MoCo v2 (He et al., 2020) self-supervised learning (SSL) objective, while Dino-MC (Wanyan et al., 2023) and Skysense (Guo et al., 2024) extend DINO (Caron et al., 2021) to multi-scale and multi-modal settings. SatMAE (Cong et al., 2022), CROMA (Fuller et al., 2024), OmniSat (Astruc et al., 2024), and DOFA (Xiong et al., 2024) apply MIM for SSL on RS imagery. Climate FMs such as ClimaX (Nguyen et al., 2023) were pretrained on multi-source climate data using MIM. GeoVLFMs such as GeoChat (Kuckreja et al., 2024) and EarthGPT (Zhang et al., 2024) are vision-language foundation models by using aligned pairs of RS images and text which are trained using common masked language model objectives and LoRA (Hu et al., 2022). However, most existing GeoFMs focus on overhead RS images and/or language modality while overlooking other geospatial data modalities, including ground-level imagery and geospatial vector data, thus lacking precise spatial reasoning and understanding capabilities. Our work addresses these issues by integrating RS imagery, SV imagery, and geolocation into the same SSL framework by using a geo-alignment-based contrastive learning objective.

**Implicit Neural Representations.** Implicit Neural Representations (INR) have been widely applied across various domains, including image regression (Tancik et al., 2020), compression (Dupont et al., 2021), 3D reconstruction (Mescheder et al., 2019), image super-resolution (Chen et al., 2021b; Gao et al., 2023), etc. The core idea behind INR is to learn a continuous function that maps spatial coordinates to corresponding signals, enabling flexible, resolution-independent data representation. A common approach is to transform spatial coordinates into multi-scale features using Fourier feature mappings (Tancik et al., 2020), which are then processed by a multi-layer perceptron (MLP) to learn a continuous representation for downstream tasks. This technique has been particularly effective in image super-resolution such as LIIF (Chen et al., 2021b) and CiaoSR (Cao et al., 2023), which directly learn pixel-wise feature mappings for image restoration. In the geospatial domain, implicit neural representations have been utilized for POI type prediction (Mai et al., 2020b), geo-aware species fine-grained recognition (Mac Aodha et al., 2019; Mai et al., 2023a;b; Sastry et al., 2025), species distribution modeling (Cole et al., 2023; Lange et al., 2023; Hamilton et al., 2024), image geolocalization (Vivanco Cepeda et al., 2024; Wang et al., 2025), satellite image classification and regression (Klemmer et al., 2023; Wu et al., 2024), and geographic question answering (Mai et al., 2020a; Li et al., 2023b; 2022b). In this work, a novel implicit neural representation module is proposed to refine the RS representations $f(r_i)$ into a localized RS embedding $\hat{r}_i(x)$ that is geographically aligned with SV image embedding $g(s_i)$ and the location embedding $e(x_i)$. Three embeddings – $\hat{r}_i(x)$, $g(s_i)$, and $e(x_i)$ – are trained in a self-supervised manner through contrastive learning.

## 3 METHODS

### 3.1 FACTORIZED ENCODER FOR GEOSPATIAL MODALITIES

We define an unlabeled geo-tagged image dataset as $X = \{(r_i, s_i, x_i) \mid i = 1, \ldots, M\}$, where $r_i$ is a remote sensing image, $s_i$ a street view image, and $x_i$ the location (longitude and latitude) of $s_i$. Here, $x_i$ is within the spatial footprint $A(r_i)$ of $r_i$ but might not be the geometric center or pixel/patch center of $r_i$. Inspired by recent contrastive pretraining models (Radford et al., 2021; Mai et al., 2023a; Guo et al., 2024), we adopt a factorized encoder architecture to extract modality-specific features independently, as shown in Figure 2. Specifically, we introduce a remote sensing image encoder $f(\cdot)$, a street view image encoder $g(\cdot)$, and a location encoder $e(\cdot)$. This modular design allows each encoder to capture unique spatial and semantic characteristics with the pretrained encoder of that modality, while forming the basis of **multimodal geographically alignment** in a later stage.

**The location encoder** $e(\cdot)$ is defined as a function $e_\theta(\mathbf{x}_i) : \mathbb{S}^2 \to \mathbb{R}^d$, parameterized by $\theta$, which maps any coordinate $\mathbf{x}_i = (\lambda_i, \phi_i)$ on the spherical surface $\mathbb{S}^2$ to a $d$-dimensional vector representation.

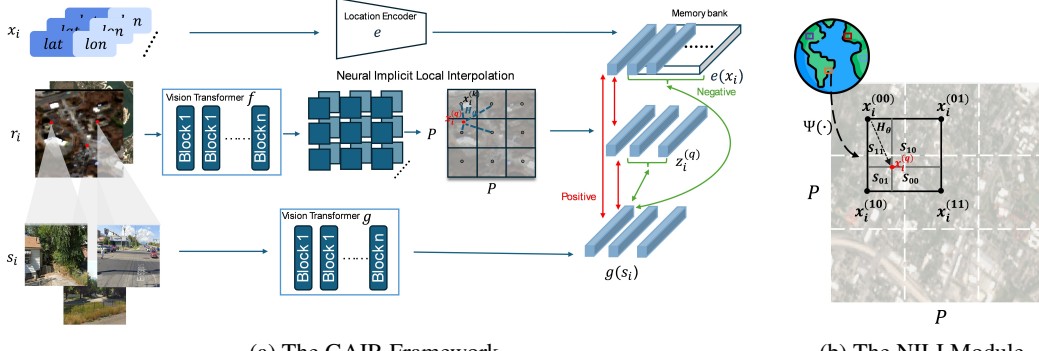

(a) The GAIR Framework          (b) The NILI Module

Figure 2: **(a) GAIR architecture:** three encoders map geolocation $x_i$, street view image $s_i$, and RS image $r_i$ into embeddings. A Neural Implicit Local Interpolation (NILI) module refines $f(r_i)$ into a localized embedding $\hat{z}_i$ at $x_i$, and contrastive learning aligns all modalities. **(b) NILI illustration:** for a query location $x_i^{(q)}$, four neighboring patch embeddings $z_i^{(k)}$ are used by a shared MLP $H_\theta$ with coordinate offsets. The weighted sum of predictions yields the final embedding $\hat{z}_i$.

Here, the longitude $\lambda_i \in [-\pi, \pi)$ and latitude $\phi_i \in [-\pi/2, \pi/2]$. We leverage existing 2D location encoders (Wu et al., 2024), specifically the Random Fourier Features (RFF) implemented from GeoCLIP (Vivanco Cepeda et al., 2024), due to its strong performance in prior works. The location encoding is represented as $e(x_i)$. Accurate geolocation plays a critical role in enabling cross-modal alignment. If the coordinates are perturbed (e.g., GPS noise), the alignment between modalities becomes less reliable and th emodel performance degrades. We analyze this effect in Appendix A.6.

**The remote sensing image encoder** $f(\cdot)$ is deployed as a Vision Transformer (ViT) (Dosovitskiy, 2020) due to its remarkable performance and generalizability across diverse computer vision tasks in recent vision foundation works (Cong et al., 2022; Fuller et al., 2024; Guo et al., 2024). Motivated by this, we employ a ViT as the RS image encoder $f(\cdot)$. For stable training, $f(\cdot)$ is initialized with a pretrained RSFM checkpoint. To retain a rich spatial representation for the follow-up neural implicit local interpolation operation, the output $f(r_i)$ is a patch-wise feature map, i.e., $f(r_i) \in \mathbb{R}^{P \times P \times D}$, where $P^2$ is the number of patches in the ViT backbone, preserving the spatial structure of $r_i$.

**The street view image encoder** $g(\cdot)$ is also a ViT model. Since we do not need the path-level representation, we directly perform an average pooling on the patch-wise feature map to output a global image-level embedding $g(s_i)$.

## 3.2 Neural Implicit Local Interpolation

Since $s_i$'s location $x_i$ might not be at the geometric center of $r_i$ or any of $r_i$'s patch/pixel, $g(s_i)$ and $f(r_i)$ are not directly geographically aligned for the contrastive learning purpose. Thus, we need to perform *spatial interpolation* based on $f(r_i)$ at location $x_i \in A(r_i)$ to produce a new geo-aligned localized RS image embedding $z_i^{(q)}$. To do that, we propose a special INR module $\Omega(\cdot)$ called **neural implicit local interpolation** which can interpolate and extract a **detailed localized RS representation** at any location $x \in A(r_i)$.

Formally, given an overhead RS image $r_i$, our RS image encoder $f(\cdot)$ extract an image representation $f(r_i) \in \mathbb{R}^{P \times P \times D}$. Each patch embedding $z_i^{(k)} \in \mathbb{R}^D$ corresponds to a real-world geographic location $x_i^{(k)} \in A(r_i)$ as the geometric center of this image patch, where $k$ is an index for any image patches which will be used in our $\Omega(\cdot)$ function.

To extract a localized RS image embedding at the location $x_i$ of an SV image $s_i$, we develop a novel INR module $\Omega(\cdot)$ on top of the RS image representation $f(r_i)$ inspired by LIIF (Chen et al., 2021b). $\Omega(\cdot)$ can learn a continuous representation of an RS image across space and be able to extract image embeddings at any query geolocation $x_i^{(q)} \in A(r_i)$. Instead of directly retrieving a single latent code $z_i^{(q)}$ at coordinate $x_i^{(q)}$ by using grid sample function (Jaderberg et al., 2015), we generate $z_i^{(q)}$ by interpolating from the four nearest patch embeddings $z_i^{(k)}$ (top-left, top-right, bottom-left,

bottom-right) of location $x_i^{(q)}$ to ensure smooth transitions across space:

$$z_i^{(q)} = \Omega(f(r_i), x_i^{(q)}) = \sum_{k \in \{00,01,10,11\}} \frac{S_k}{S} \cdot H_\theta(z_i^{(k)}, \Psi(x_i^{(q)}) - \Psi(x_i^{(k)})) \quad (1)$$

Here, $z_i^{(k)} \in f(r_i)$ ($k \in \{00, 01, 10, 11\}$) denote the four nearest image patch embeddings of query location $x_i^{(q)}$. $x_i^{(k)}$ denotes the geographic coordinates of $z_i^{(k)}$. $S_k$ is the area of the rectangle formed between $x_i^{(k)}$ and $x_i^q$. $\Psi(\cdot)$ is a projection function that transforms the geographic coordinates (e.g., $x_i^{(q)}$ and $x_i^{(k)}$) into the image coordinate space. $H_\theta$ is a multilayer perception (MLP) modulated by the latent code $z_i^{(k)}$ which takes the projected coordinate difference between $\Psi(x_i^{(q)})$ and $\Psi(x_i^{(k)})$ and predict the localized RS embedding at location $x_i^q$. The final localized RS embedding $z_i^{(q)}$ is computed as the *weighted sum* of the four independent predictions based on $S_k/S$ where $S = \sum_k S_k$. This approach ensures local feature continuity by enabling overlapping representations from neighboring latent codes. At each query location, four independent predictions are ensembled, leading to smooth and spatially coherent feature synthesis. Our NILI module $\Omega(\cdot)$ is agnostic to the implementation of $f(\cdot)$, which can be ViT- or CNN-based encoders as long as they can produce a 2D image feature map $f(r_i) \in \mathbb{R}^{P \times P \times D}$ for spatial interpolation. Please refer to Figure 2b and Appendix A.2 for details.

### 3.3 GEO-ALIGNED CONTRASTIVE OBJECTIVE

Given $e(x_i)$, $z_i^{(q)}$, and $g(s_i)$, we can leverage their geospatial relationships to form SSL objectives to learn generalizable neural representations. Here, we mainly use two contrastive learning objectives:

**Implicit Neural Contrastive Learning (INCL).** The key idea of INCL is performing contrastive learning between two geo-aligned image representations – localized RS embedding $z_i^{(q)}$ at $x_i^{(q)}$ and its corresponding SV embedding $g(s_i)$:

$$\mathcal{L}_{\text{INCL}} = -\frac{1}{2N} \sum_{i=1}^{N} \left[ \log \frac{\exp(\mathcal{D}(z_i^{(q)}, g(s_i))/\tau)}{\sum\limits_{j=1}^{N} \exp(\mathcal{D}(z_i^{(q)}, g(s_j))/\tau)} + \log \frac{\exp(\mathcal{D}(g(s_i), z_i^{(q)})/\tau)}{\sum\limits_{j=1}^{N} \exp(\mathcal{D}(g(s_i), z_j^{(q)})/\tau)} \right] \quad (2)$$

where $\mathcal{D}(\cdot, \cdot)$ denotes a cosine similarity function, $N$ is the batch size, and $\tau$ is a temperature parameter. This contrastive loss enforces positive pairs between the matched embeddings while discouraging alignment with non-matching locations (Radford et al., 2021). The first term ensures that the extracted localized RS embedding $z_i^{(q)}$ is aligned with the corresponding co-located SV embedding $g(s_i)$, while the second term enforces the inverse alignment, treating $g(s_i)$ as the anchor.

**Spatially Explicit Contrastive Learning (SECL).** To further reinforce the geospatial consistency across data modalities, we introduce SECL, which incorporates explicit location encoding $e(x_i)$ into the contrastive learning framework. We construct a memory bank $\mathcal{M}$ (He et al., 2020; Vivanco Cepeda et al., 2024) to store location embeddings $e(x_i)$ from past mini-batches. The SECL objective consists of two separate contrastive losses: one aligning location embeddings with RS embeddings, and another aligning location embeddings with SV embeddings. The SECL loss is defined as:

$$\mathcal{L}_{\text{SECL}} = -\frac{1}{2N} \sum_{i=1}^{N} \left[ \log \frac{\exp(\mathcal{D}(e(x_i), z_i^{(q)})/\tau)}{\sum\limits_{j \in \mathcal{M}} \exp(\mathcal{D}(e(x_j), z_i^{(q)})/\tau)} + \log \frac{\exp(\mathcal{D}(e(x_i), g(s_i))/\tau)}{\sum\limits_{j \in \mathcal{M}} \exp(\mathcal{D}(e(x_j), g(s_i))/\tau)} \right] \quad (3)$$

Finally, GAIR's pre-training objective is the sum of two objectives, where $\lambda$ is a hyperparameter to control the contribution of SECL:

$$\mathcal{L} = \mathcal{L}_{\text{INCL}} + \lambda \mathcal{L}_{\text{SECL}} \quad (4)$$

### 3.4 TRANSFER LEARNING

To assess the performance of the pre-trained GAIR, we employ it both as a feature extractor for linear probing and as a model parameter initialization for model fine-tuning. GAIR comprises three encoders: two image encoders dedicated to remote sensing and street view images, and a location encoder responsible for location data representation.

**Fine-Tuning the StreetView Image Encoder $g(\cdot)$.** For the SV image encoder $g(\cdot)$, we remove the projection layer and introduce a new head $h_g(\cdot)$ to process the extracted SV image feature vectors. We perform different experimental setups – $h_g(\cdot)$ is a single linear layer for linear probing and two linear layers for non-linear probing.

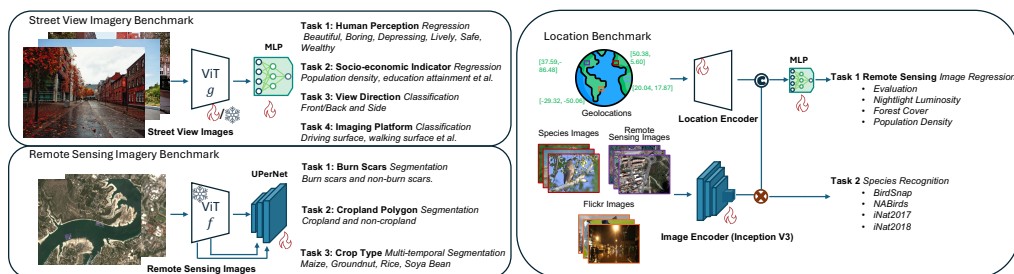

Figure 3: The evaluation pipelines of three benchmarks (9 tasks). After pretraining GAIR, we fine-tune the street view image encoder $g$ to tackle the street view imagery benchmark, the RS image encoder $f$ for remote sensing imagery benchmark, and the location encoder $e$ for location benchmark.

**Fine-Tuning the Remote Sensing Image Encoder $f(\cdot)$.** Similarly, we remove the projection layer and introduce a new head $h_f(\cdot)$. For RS image semantic segmentation tasks, we use UPerNet (Xiao et al., 2018) as $h_f(\cdot)$, while for RS image change detection, we adopt a Siamese UPerNet as $h_f(\cdot)$ to capture temporal changes. For multi-temporal datasets, we experiment with two temporal aggregation strategies: (1) a naive linear mapping, and (2) a lightweight spatial-temporal encoder (L-TAE) (Garnot & Landrieu, 2020) to better integrate temporal information.

**Fine-Tuning the Location Encoder $e(\cdot)$.** We also introduce a new head $h_e(\cdot)$ for the location encoder $e(\cdot)$. Following standard protocols for the geo-aware image classification (Mac Aodha et al., 2019; Wu et al., 2024), given a location-image pair $(x, I)$, where $I$ denotes an image, the model predicts the category $y$ by factorizing the probability as $P(y|I, x) \propto P(y|I)P(y|x)$. Here, $P(y|I)$ is estimated using an image encoder; in this paper, we utilize Inception V3 network (Szegedy et al., 2016) following the standard implementation (Mai et al., 2020b; Wu et al., 2024), while $e(x)$ contributes to $P(y|x)$. The location encoder is fine-tuned using cross-entropy loss for the geo-aware image classification tasks. For geo-aware image regression tasks, we concatenate the image embedding and location embedding and feed the result into a linear layer for regression. We also use MSE as the loss function.

## 4 EXPERIMENTS

**Pretraining.** We construct a large-scale pretraining dataset, **Streetscapes1M**, containing 1M triples $(r_i, s_i, x_i)$. Street view images are uniformly sampled from the Global Streetscapes dataset (Hou et al., 2024), and for each sampled location, we collect the corresponding Sentinel-2 remote sensing imagery from Google Earth Engine (GEE) (Gorelick et al., 2017). Following Ayush et al. (2021), we obtain monthly temporal augmentations of Sentinel-2 data (12 surface reflectance bands, with the cirrus band removed). The street view images are resized to $400 \times 400$ and RS crops to $120 \times 120$. All pretraining, fine-tuning, and ablations are conducted with ViT-B on a Linux server with 4 NVIDIA RTX A6000 GPUs. Further dataset construction and implementation details are provided in Appendix A.3 and A.8, and computation cost analysis in Appendix A.9.

**Model Fine-Tuning.** For street view imagery tasks, we adopt three fine-tuning strategies: 1) **Linear probing**, where the encoder remains frozen, and only a linear layer is trained; 2) **Non-linear probing**, where the encoder is fixed, and a two-layer MLP head with a sigmoid activation function is optimized; and 3) **Full fine-tuning**, where all model parameters are updated. For remote sensing benchmarks, we fine-tune only the UPerNet while keeping the pretrained backbone frozen. For location-based tasks, we apply full fine-tuning to optimize all parameters. A comprehensive overview of the fine-tuning pipeline, baselines, and task configurations is provided in Figure 3 and Appendix A.4.

**Benchmarks.** To comprehensively evaluate our GAIR, we conduct experiments on 9 different geospatial tasks and 22 datasets. These tasks include 4 street view image tasks – 1) Human perception regression, 2) Socio-economic indicator regression, 3) View direction classification, and 4) Imaging platform classification; 3 remote sensing tasks – 1) Burn scars segmentation, 2) Crop Type Mapping-South Sudan, and 3) Cropland polygon segmentation; 2 geolocation tasks – 1) Species recognition, and 2) Remote sensing image regression. Please refer to Figure 3 and Appendix A.7 for detailed descriptions of different tasks and datasets.

**Baselines.** To thoroughly evaluate the performance of GAIR, we select different sets of baselines for each benchmark category, ensuring a fair and comprehensive comparison.

For street view imagery benchmarks, we compare GAIR against three representative and reproducible GeoFMs: SatMAE (Cong et al., 2022), CROMA (Fuller et al., 2024), PIS (An et al., 2024) and the

Table 1: Comparisons of model performance across street view imagery benchmark. RMSE ↓ is reported for *socio-economic indicator regression and human perception regression*, while F1 score ↑ is reported for *view direction classification and imaging platform classification*. The best and second-best results are denoted as **Bold** and Underline. The detailed results are in Appendix A.12

| Model | Socio-Eco. Indic. (RMSE ↓) | | | Human Perception (RMSE ↓) | | | View Direction (F1 ↑) | | | Imaging Platform (F1 ↑) | | |
|---|---|---|---|---|---|---|---|---|---|---|---|---|
| | Linear | Non-Linear | All | Linear | Non-Linear | All | Linear | Non-Linear | All | Linear | Non-Linear | All |
| SatMAE(Cong et al., 2022) | 0.9671 | 0.9421 | 0.7690 | 2.1681 | 2.1196 | 1.8812 | 0.0000 | 0.0202 | 0.3885 | 0.1916 | 0.1596 | 0.2114 |
| CROMA(Fuller et al., 2024) | 0.9550 | 0.9295 | 0.7927 | 2.0768 | 2.0219 | 1.9387 | 0.0064 | 0.0640 | 0.2133 | 0.1733 | 0.1832 | 0.2369 |
| PIS(An et al., 2024) | 0.9290 | 0.8911 | 0.6797 | 1.9858 | 1.8865 | 1.6166 | 0.1937 | 0.2442 | 0.3537 | 0.2315 | 0.2796 | 0.3328 |
| TaxaBind(Sastry et al., 2025) | 0.8819 | **0.8296** | 0.6654 | 1.7960 | 1.7108 | 1.6067 | 0.2617 | 0.3971 | 0.5088 | 0.2323 | 0.2541 | 0.3052 |
| Random Init. | 0.9700 | 0.9524 | 0.9035 | 2.2000 | 2.1534 | 2.1839 | 0.0000 | 0.0000 | 0.3954 | 0.1834 | 0.1895 | 0.1970 |
| ImageNet Init.(Wu et al., 2020) | 0.8929 | 0.8489 | 0.7012 | 1.7473 | **1.6439** | 1.5271 | 0.5328 | 0.4186 | 0.5850 | 0.3229 | 0.3275 | 0.3540 |
| MoCo v3(Chen et al., 2021a) | **0.8760** | 0.8441 | 0.6779 | 1.7590 | 1.6599 | 1.5821 | 0.2106 | 0.6262 | 0.4976 | 0.2584 | 0.3171 | 0.3161 |
| MAE-ImageNet(He et al., 2022) | 0.8816 | 0.8373 | 0.7150 | 1.8730 | 1.7592 | 1.5788 | 0.4298 | 0.6426 | 0.4961 | 0.2619 | 0.3257 | 0.2195 |
| MoCo v3-Streetscapes | 0.8863 | 0.8429 | 0.6707 | 1.7601 | 1.7084 | 1.5800 | 0.3508 | 0.5988 | 0.5032 | 0.3439 | 0.3308 | 0.3639 |
| MAE-Streetscapes | 0.8940 | 0.8416 | 0.6872 | 1.8514 | 1.7701 | 1.5774 | 0.3451 | 0.6102 | 0.4921 | 0.3335 | 0.3482 | 0.3050 |
| GAIR-MAE | 0.9485 | 0.9011 | 0.8048 | 2.1254 | 2.0495 | 1.9611 | 0.0272 | 0.0895 | 0.3029 | 0.2124 | 0.2582 | 0.3249 |
| GAIR w/o Loc | 0.8823 | 0.8352 | 0.6725 | 1.7473 | 1.6782 | 1.5960 | 0.5290 | 0.6052 | 0.6078 | 0.3676 | 0.3709 | 0.3892 |
| GAIR | 0.8803 | 0.8349 | **0.6612** | **1.7141** | 1.6489 | **1.5072** | **0.5457** | **0.6495** | **0.6102** | **0.3793** | **0.3782** | **0.4071** |

ground image encoder from TaxaBind (Sastry et al., 2025). To benchmark against general-purpose vision models, we also include several Vision Transformers (ViTs) with different initializations, including randomly initialized weights (Random Init.), supervised pretrained on ImageNet (ImageNet Init.) (Deng et al., 2009; Wu et al., 2020),pretrained on ImageNet using MoCo-v3 (MoCo v3-ImageNet) (Chen et al., 2021a), and pretrained using MAE-based objective (MAE-ImageNet) (He et al., 2022). Finally, to specifically analyze the benefits of domain-adapted pretraining, we train two models, MoCo-v3-Streetscapes and MAE-Streetscapes, on our Streetscapes1M dataset.

For RS imagery benchmarks, we adopt PANGAEA-Bench (Marsocci et al., 2024), which includes 12 GeoFM baselines. We refer to their work for detailed settings. In addition, we also include the TaxaBind satellite encoder Sastry et al. (2025). For location benchmarks, we employ the LocBench from TorchSpatial (Wu et al., 2024) to evaluate the effectiveness of different initialization strategies for RFF. We compare three initialization methods: 1) Random Init., where the model is trained from scratch, 2) GeoCLIP Init., which leverages a pretrained checkpoint from GeoCLIP (Vivanco Cepeda et al., 2024), 3) TaxaBind Init., which uses the location encoder in TaxaBind. , and 4) GAIR Init., initialized by our pretrained checkpoint. Additionally, we include a No Prior baseline, which uses only image features and omits location embeddings entirely.

**Ablation Study.** We conduct ablation studies to analyze the impact of different components in GAIR, including two variants (see Appendix A.5 for detailed illustrations and results):

- **GAIR-MAE** – In this setting, we replace GAIR's contrastive learning objective with a masked autoencoder (MAE) objective (He et al., 2022). Specifically, the street view image is treated as a special masked patch of the RS image, allowing the model to reconstruct missing image patches.
- **GAIR w/o Loc** – To assess the significance of geolocation encoding, we remove the location embedding $e(x_i)$ from GAIR.

### 4.1 STREET VIEW IMAGERY RESULTS

**Socio-economic Indicator Regression.** This task aims at predicting socio-economic indicators from street view images, a widely studied problem in urban analytics (Fan et al., 2023). We evaluate models on 10 socio-economic indicators: population density, educational attainment, health condition rates, racial demographics, median household income, proximity to public transportation, percentage of people who walk or bike, proportion of the population over 65 years old, crime rates, and visible sky area, and report RMSE across different fine-tuning settings in Table 1. GAIR achieves the lowest RMSE in the Fine-Tune All settings and the 2nd best in Non-Linear Probing settings.

**Human Perception Regression.** This task assesses human perceptions of urban environments, which are widely used in urban planning, psychology, and social studies (Wei et al., 2022). We perform regression on six perceptual attributes: 'Beautiful', 'Boring', 'Depressing', 'Lively', 'Safe', and 'Wealthy'. As shown in Table 1, we can see that GAIR achieves the best performance on linear probing and fine-tuning settings, and remains the 2nd best model on the non-linear probing setting. Notably, the explicit integration of geolocation into contrastive learning significantly enhances model performance, as seen in the substantial improvement from GAIR w/o Loc to GAIR.

**View Direction Classification.** View direction reflects the model's ability to capture geospatial context. This task involves classifying images into two categories: "front/back" and "side". As shown

in Table 1, existing GeoFMs struggle with view direction estimation, whereas GAIR consistently outperforms all baselines, demonstrating its superior spatial awareness.

**Imaging Platform Classification.**    Different imaging platforms offer distinct perspectives on the built and natural environment. For this task, we include 6 platform types, including driving surface, walking surface, cycling surface, tuner, open fields, and railway. As shown in Table 1, GAIR outperforms all baselines on all settings.

### 4.2   REMOTE SENSING RESULTS

Table 2 presents the evaluation results of GAIR across three remote sensing tasks.

**Single Temporal Semantic Segmentation.**    We evaluate GAIR on two single-temporal segmentation tasks: burn scar segmentation using the HLS Burns dataset (Jakubik et al., 2023) and cropland polygon delineation using the AI4SmallFarms dataset (Persello et al., 2023), both based on Sentinel-2 imagery. GAIR achieves a mean Intersection over Union (mIoU) of 83.26% for burn scars and 43.35% for cropland segmentation, outperforming all baselines. Notably, GAIR improves performance by 0.5% for burn scars and 16.16% for cropland segmentation, demonstrating its capability in extracting meaningful geospatial features for land cover classification.

Table 2: Performance comparison (mIoU ↑) of GAIR and other Geo-Foundation Models (GeoFMs) across four remote sensing semantic segmentation (SS) tasks. Linear and L-TAE (Garnot & Landrieu, 2020) represent two different multi-temporal augmentation strategies

| Task | Single Temp. SS | | Multi Temp. SS | |
|---|---|---|---|---|
| | Burn Scars | Crop. Poly. | Crop Type | |
| Model | | | Linear | L-TAE |
| CROMA (Fuller et al., 2024) | 81.95 | 25.65 | 47.02 | 49.38 |
| DOFA (Xiong et al., 2024) | 78.96 | 27.07 | 49.81 | 51.33 |
| GFM-Swin (Mendieta et al., 2023) | 76.17 | 27.19 | 39.72 | 46.98 |
| Prithvi (Jakubik et al., 2023) | 82.67 | 26.86 | 39.92 | 43.07 |
| RemoteCLIP (Liu et al., 2024) | 75.55 | 25.12 | 46.50 | 52.05 |
| SatlasNet (Bastani et al., 2023) | 79.69 | 25.13 | 46.97 | 46.97 |
| Scale-MAE (Reed et al., 2023) | 76.71 | 21.47 | 21.39 | 25.42 |
| SpectralGPT (Hong et al., 2024) | 80.47 | 26.75 | 53.50 | 46.95 |
| TaxaBind (Sastry et al., 2025) | 75.84 | 38.46 | 44.80 | 43.52 |
| S12-Data2Vec (Stewart et al., 2023) | 81.14 | 24.23 | 54.01 | _54.03_ |
| S12-DINO (Stewart et al., 2023) | 81.44 | 25.62 | 46.56 | 48.66 |
| S12-MAE (Stewart et al., 2023) | 80.86 | 24.69 | 46.28 | 45.80 |
| S12-MoCo (Stewart et al., 2023) | 80.76 | 25.38 | 44.22 | 48.58 |
| GAIR-MAE | 74.15 | 22.77 | 34.18 | 40.44 |
| GAIR w/o Loc | _82.94_ | _43.28_ | _55.41_ | **54.32** |
| GAIR | **83.26** | **43.35** | **55.53** | 54.01 |

**Multi Temporal Semantic Segmentation.**    For this task, we utilize the crop type mapping dataset (M Rustowicz et al., 2019), which consists of Sentinel-2 imagery from 2017. To utilize temporal information, we employ two widely used feature aggregation strategies: linear aggregation and L-TAE aggregation. GAIR consistently outperforms other baselines, particularly in the simpler linear aggregation setting, achieving a 1% mIoU improvement.

Table 3: The location benchmark results (LocBench) for image regression and species recognition. The "PopDen", "ForCov", "NightLum", and "Elev." columns indicate regression prediction results on population density, forest coverage, nightlight luminosity, and elevation.

| Init. | Model | Image Regression ($R^2$ ↑) | | | | Species Recognition (Top-1 accuracy ↑) | | | |
|---|---|---|---|---|---|---|---|---|---|
| | | PopDen | ForCov | NightLum | Elev. | BirdSnap | NABirds | iNat17 | iNat18 |
| Rand (Wu et al., 2024) | No Prior | 0.38 | 0.52 | 0.33 | 0.27 | 70.07 | 76.08 | 63.27 | 60.20 |
| | RFF | 0.57 | _0.84_ | 0.35 | 0.76 | 70.07 | 81.63 | 67.73 | 71.66 |
| GeoCLIP (Vivanco Cepeda et al., 2024) | RFF | _0.61_ | _0.84_ | 0.37 | 0.78 | 70.56 | 81.65 | 67.78 | 71.93 |
| TaxaBind (Sastry et al., 2025) | RFF | 0.60 | 0.77 | 0.38 | 0.72 | **72.15** | 80.58 | **68.18** | 71.71 |
| GAIR | RFF | **0.67** | **0.86** | _0.40_ | **0.82** | _72.07_ | _81.76_ | _67.84_ | **72.48** |
| GAIR *Debias* | RFF | **0.67** | _0.84_ | **0.41** | _0.81_ | 72.03 | **81.88** | _67.84_ | _72.17_ |

### 4.3   LOCATION BENCHMARK RESULTS

Here, we evaluate 2 location tasks. Table 3 shows all the experimental results.

**Geo-Aware Image Regression.**    We evaluate the effectiveness of location priors using the datasets from MOSAIKS (Rolf et al., 2021), which includes four regression tasks: population density, forest cover, nightlight luminosity, and elevation estimation. The results indicate that incorporating location information significantly enhances model performance, yielding up to a 50% improvement over models without location priors. Furthermore, using pretrained checkpoints further refines the learned geospatial representations. In particular, pretraining with GAIR provides the greatest gains, achieving improvements of 0.06, 0.02, 0.03, and 0.04 in $R^2$ for population density, forest cover, nightlight luminosity, and elevation regression, respectively.

**Geo-Aware Species Recognition.**    This task aims to classify images of different animal species. We use four datasets: BirdSnap (Berg et al., 2014), NABirds (Van Horn et al., 2015), iNat2017 (Van Horn et al., 2018), and iNat2018 (Van Horn et al., 2018). Using a pretrained GeoCLIP encoder provides a moderate performance boost over random initialization. Notably, GAIR achieves the best results on NABirds and iNat2018 while remaining strongly competitive on BirdSnap and iNat2017 with TaxaBind. Notably, GAIR is pretrained on our Streetscapes1M dataset that does not contain any

species images but streetview images, whereas TaxaBind (Sastry et al., 2025) was pretrained on 2.55 million colocated satellite and ground-level species image pairs, giving TaxaBind a strong advantage for this task. These results clearly demonstrate the effectiveness of GAIR's pretraining strategy and **strong out-of-domain generalizability**.

## 5 DISCUSSIONS

**Multi-Modal Pretraining.** GAIR uses geo-aligned contrastive learning across three modalities, yielding notable gains even in single-modal tasks. As shown in Table 1, it outperforms MoCo v3 and MAE trained on Streetscapes1M by up to 0.1 RMSE and 0.2 F1 score. Despite using only 0.1M RS images (Appendix A.10), GAIR achieves strong results in RS tasks, surpassing GeoFMs trained on much larger datasets (Table 2). Multimodal fusion further improves performance by 4–20% (Appendix A.13)

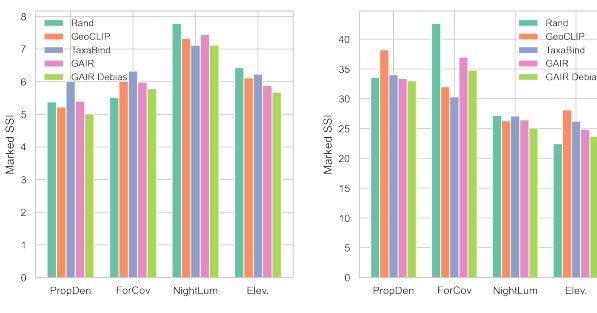

(a) Image Regression     (b) Species Recognition

Figure 4: Geo-Bias Score on the LocBench.

**Model Debias.** Street view imagery is biased toward urban regions, as rural areas have limited coverage. Thus, pretraining GAIR exclusively on Streetscapes1M risks amplifying geographic bias. In order to debias GAIR across space, we further collected urban-rural balanced RS–Loc pairs to continue pretraining GAIR's RS and location encoders. We denote the resulting model as **GAIR *Debias***. From Table 3 and Figure 4, we can see that this approach notably reduces geographic bias as measured by the Geo-Bias Score (Wu et al., 2024) without sacrificing model performance. Details are provided in Appendix A.14.

### 5.1 ANALYSIS OF MULTIMODAL GEO-ALIGNMENT

We evaluate whether GAIR captures spatial relations across modalities by computing cosine similarities between an SV image embedding $g(s_i) \in \mathbb{R}^D$ and different localized RS image embedding $z_i^{(q)}$ generated by $\Omega$ at different locations $x_i^{(q)}$. As shown in Figure 5(a), GAIR successfully learns to geographically align these two representations. We further compare $g(s_i) \in \mathbb{R}^D$ with location embeddings $e(x_i)$ sampled on a uniform mesh around the image location, and observe consistent alignment with street-level features (Figure 5(b)). See Appendix A.16 for more visual examples. By leveraging GAIR's geo-alignment across modalities, we also conduct an SV image geolocalization experiment. Results show that GAIR can outperform GeoCLIP (Vivanco Cepeda et al., 2024) across all distance thresholds. Please see Appendix A.15 for detailed quantitative results.

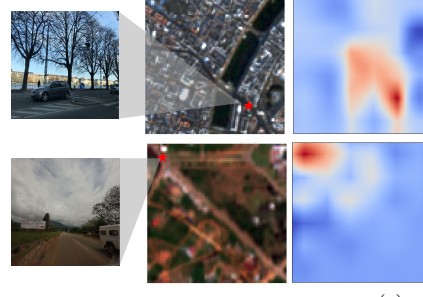

(a) Alignment between $g(s_i)$ and $z_i^{(q)}$

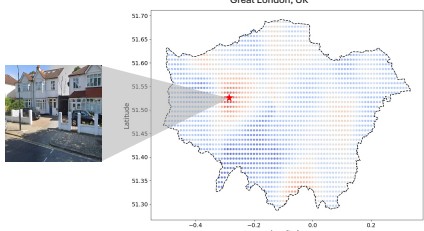

(b) Alignment between $g(s_i)$ and $e(x_i)$

Figure 5: Visualization of cross-modal alignment in GAIR via heat maps. Red stars mark the SV image location $x_i$. (a) Cosine similarity between $g(s_i)$ and localized RS embeddings $\hat{z}_i$. (b) Cosine similarity between $g(s_i)$ and location embeddings $e(x_i)$ sampled across Greater London.

## 6 CONCLUSION

We introduce GAIR, a new GeoFM that integrates remote sensing imagery, street view imagery, and geolocation data to enhance geospatial representation learning. By using neural implicit local interpolation, GAIR explicitly aligns different representations geographically whose effectiveness has been shown in 9 downstream tasks and 22 datasets. However, GAIR does not consider view direction information (e.g., SV v.s. RS images), which should be addressed in future work.

## REPRODUCIBILITY STATEMENT

We provide detailed descriptions of model architecture, objectives, datasets, and hyperparameters in Section 3 and 4 and Appendix A.3, A.7 and A.8. To support replication, we have uploaded anonymized source code as supplementary materials.

## ETHICS STATEMENT

We use only publicly available datasets: street-view images paired with Sentinel-2 from GEE for pretraining, and established benchmarks for evaluation; experiments operate at regional/task level rather than individual profiling. All datasets are used under their licenses, and results are reported for scientific benchmarking only. We adhere to the ICLR Code of Ethics throughout submission, review, and discussion.

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

# A APPENDIX

## A.1 LLM USAGE

Large Language Models (LLMs) are used only to polish the writing of this paper, for example improving grammar and phrasing. LLMs are not involved in research idea exploration, methodology design, data analysis, or experimental results. All technical content and claims are originally written by the authors.

## A.2 NEURAL IMPLICIT LOCAL INTERPOLATION

We parameterize the continuous function with a small neural network $H_\theta$ (a multilayer perceptron). This implicit decoder $H_\theta$ is shared across all RS images and takes two inputs: (a) a latent code (patch embedding) $z$, and (b) a continuous 2D coordinate offset $\Delta \mathbf{x}$ within that latent code's cell. If we only encode the latent code by using the nearest patch embedding to the query location $x_i^q$, this could lead to discontinuities at cell boundaries (Chen et al., 2021b). To ensure smooth transitions across space, we use neural implicit local interpolation. Instead of relying on a single latent code, we query the four nearest image patches and get their latent codes $z_i^{(k)}$ ($k \in \{00, 01, 10, 11\}$) corresponding to the cell corners surrounding the continuous coordinate. Each code produces a prediction $H_\theta(z_i^{(k)}, \Psi(x_i^{(q)}) - \Psi(x_i^{(k)}))$, and we blend these outputs with weights proportional to the area of the cell corner (or equivalently, bilinear interpolation weights based on the coordinate's fractional position within the cell) (see Equation 1).

## A.3 STREETSCAPES1M DATASET CONSTRUCTION DETAILS

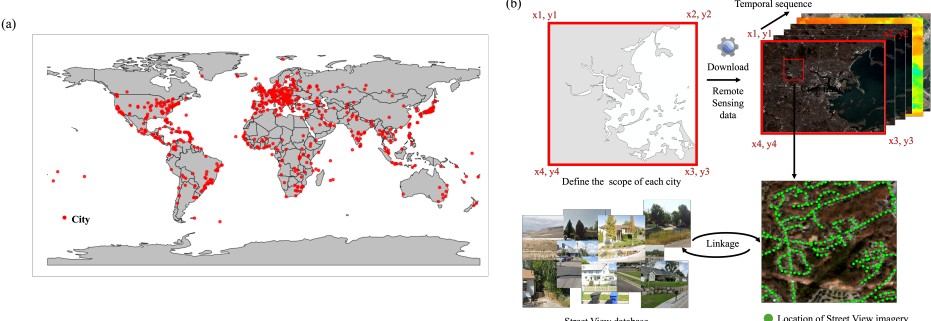

Figure 6: An illustration on the pipeline to construct our **Streetscapes1M** GeoFM pretraining dataset. (a) The geographic distributions of cities we used to construct our **Streetscapes1M** dataset. (b) The pipeline we use to construct the **Streetscapes1M** dataset.

In order to construct a global scale geospatial aligned multimodal dataset for GAIR model pretraining, we collect a dataset called **Streetscapes1M** based on the Global Streetscapes dataset (Hou et al., 2024). Figure 6 illustrates the dataset construction process of **Streetscapes1M** which can be divided into three steps:

1. **Study Area Selection:** As shown in Figure 6 (a), we sample 1 million street view images in the Global Streetscapes dataset across 688 cities globally.

2. **Remote Sensing Image Data Collection:** For each city, as shown in Figure 6 (b), we define the spatial scope and temporal scope of the study area and download Sentinel-2 multispectral remote sensing images based on the defined spatial and temporal scope. We download Sentinel-2 imagery from Google Earth Engine with the following criteria: (i) Level-2A (L2A) atmospherically corrected products, (ii) maximum cloud cover $< 20\%$, and (iii) spatial resolution of 10m for all selected bands. Following prior RS foundation models such as SatMAE (Cong et al., 2022) and CROMA (Fuller et al., 2024), we exclude atmospheric bands (e.g., cirrus, water vapor) to reduce noise. For each location in our pretraining set, we retrieve a time series of monthly images that satisfy the above conditions,

and randomly select one timestamp during training as temporal augmentation. This ensures diversity in surface conditions while controlling for cloud contamination.

3. **Geospatial Alignment Among Data Modalities:** As shown in Figure 6 (b), based on the spatial footprints of the collected overhead RS images and street view images, we establish their geospatial alignment between them and make them ready for GAIR model pre-training.

## A.4 BASELINES

To comprehensively evaluate GAIR, for the street view image encoder, we compare it against multiple baselines spanning geo-foundation models, general vision models, and self-supervised learning frameworks. Our baseline models are categorized as follows:

**Geo-Foundation Models.** These models are specifically designed for geospatial image prediction, leveraging pretraining strategies tailored to remote sensing and multimodal spatial data.

- **SatMAE** (Cong et al., 2022) – A transformer-based masked autoencoder pre-trained on large-scale remote sensing imagery, leveraging self-supervised learning to extract generalized spatial features.
- **CROMA** (Fuller et al., 2024) – A contrastive learning model designed for multimodal geospatial representation learning, incorporating spatially-aware feature alignment.
- **PIS** (An et al., 2024) – A geospatial self-supervised pretraining approach that integrates intra-instance similarity.
- **TaxaBind** (Sastry et al., 2025) – A multimodal framework that unifies six ecological modalities using species images as the binding modality, with multimodal patching enabling zero-shot and cross-modal ecological tasks.

**General Vision Models.** These models serve as broad benchmarks by assessing the performance of standard vision pretraining techniques when applied to geospatial tasks.

- **ImageNet Initialization** – ViT pretrained on ImageNet using standard self-supervised training.
- **Random Initialization** – A control setting where models are trained from scratch without any pretraining.

**Self-Supervised Learning Models.** These models leverage contrastive and masked image pre-training paradigms to learn generalizable representations without labeled supervision.

- **MoCo v3-ImageNet** – A ViT trained on ImageNet using MoCo v3.
- **MAE-ImageNet** – A ViT trained on ImageNet using a masked autoencoder (MAE) objective (He et al., 2022).
- **MoCo v3-Streetscapes** – A ViT model pre-trained on Global Streetscapes data, capturing scene-level geospatial information using MoCo v3.
- **MAE-Streetscapes** – A ViT model pre-trained on Global Streetscapes using MAE.

## A.5 ABLATION STUDIES ON GAIR

To further analyze the contribution of key components in GAIR, we design four ablation settings, results are shown in Table 4:

- **GAIR-MAE** – In this setup, we adopt a Masked Autoencoder (MAE) pretraining strategy (He et al., 2022) in place of GAIR's contrastive learning objective. Here, the model learns to reconstruct missing image patches by treating the street view image as a masked region of the remote sensing image. Figure 7 provides a detailed visualization of this approach.
- **GAIR w/o Loc** – To assess the significance of geolocation encoding, we remove the location embedding $e(x_i)$ from GAIR.

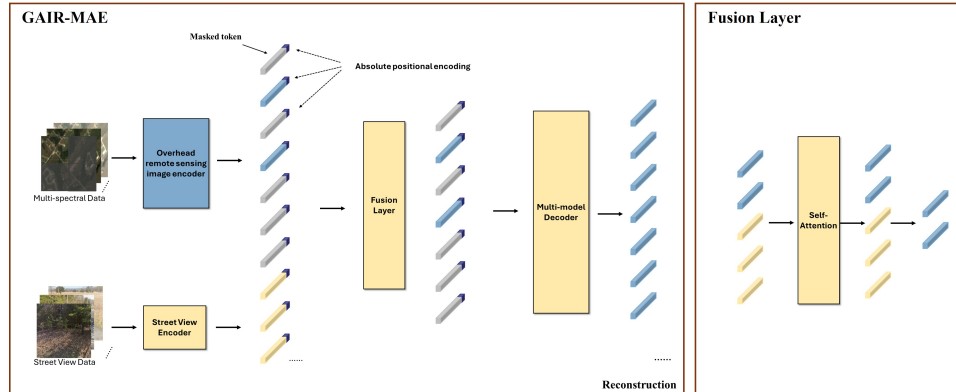

Figure 7: The architecture of one ablation setting, GAIR-MAE. In this variant, feature representations are extracted independently using two encoders, producing separate patch embeddings for remote sensing and street view modalities. Modality fusion is performed via a self-attention mechanism applied to both sets of patch embeddings. Finally, a masked autoencoder (MAE) decoder processes the fused representation to compute the reconstruction loss between the reconstructed masked regions and the input remote sensing images.

- **GAIR w/o NILI**: This variant removes the neural implicit representation (INR) module for spatially aligning remote sensing features with street view images. Instead, we apply global average pooling over the feature map produced by the remote sensing encoder $f(r_i)$ to obtain a single latent vector. This representation is then used directly for contrastive learning without the geospatial alignment step.

- **GAIR w/o NILI and Loc**: In this setting, both the INR module and the location encoder branch are removed. The model performs contrastive learning only between remote sensing and street view embeddings. This baseline is conceptually similar to (Huynh et al., 2025). The difference is that (Huynh et al., 2025) used the embeddings of species images instead of streetview images to contrast with the remote sensing image embeddings.

Table 4: Ablation study results on Street View and Remote Sensing benchmarks.

| Ablation Settings | Street View Imagery Benchmark Socio-economic Regression (RMSE ↓) | Remote Sensing Benchmark Crop Type Classification (mIoU ↑) |
|---|---|---|
| GAIR-MAE | 0.9011 | 34.18 |
| GAIR w/o Loc | 0.8352 | 55.41 |
| GAIR w/o NILI | 0.8588 | 55.01 |
| GAIR w/o NILI and Loc (Huynh et al., 2025) | 0.8451 | 52.78 |
| **GAIR** | **0.8349** | **55.53** |

Table 4 compares GAIR with all its variants on two representative tasks – streetview image-based socio-economic regression and RS image-based crop type classification. We can see that deleting some components of GAIR (e.g., INR module, location encoding module, etc.) or switching to other objectives (e.g., MAE) will lead to performance degradation on both the SV image encoder and RS image encoder. We notice that the performance of *GAIR-MAE* is significantly worse than other variants while *GAIR w/o Loc* are usually the second best. In the following, we will use both GAIR variants in other ablation studies and evaluations.

## A.6   IMPACT OF GEOLOCATION NOISE ON GEO-ALIGNMENT

A central idea of GAIR is that accurate geolocation provides the spatial anchor for aligning heterogeneous geospatial modalities. If the location metadata is corrupted, the alignment between SV and RS embeddings becomes unreliable, which should directly affect downstream task performance. To verify this, we introduce controlled noise into the GPS coordinates and measure the degradation.

Table 5: Effect of geolocation noise on GAIR for the socio-economic regression tasks. A lower RMSE reflects better performance.

| Noise Level | No noise | $\kappa = 300$ | $\kappa = 100$ | $\kappa = 1$ |
|---|---|---|---|---|
| RMSE | 0.8349 | 0.8360 | 0.8488 | 0.8517 |

**Noise Injection.**    We use the von Mises–Fisher (vMF) distribution to simulate the location noise, because it is the standard choice for modeling random perturbations on a spherical surface. The concentration parameter $\kappa$ controls the noise level: a larger $\kappa$ means less GPS noise (a tighter vMF distribution), while a smaller $\kappa$ injects stronger positional errors. Then, starting from a pretrained checkpoint (100 epochs) of GAIR, we continue pretraining GAIR for 2 additional epochs with noisy coordinates, which are derived by adding a GPS noise to the ground truth locations drawn from vMF distributions with different $\kappa$ values. We then evaluate GAIR trained on noisy locations with different noise levels on the socio-economic indicator regression benchmark using non-linear probing.

**Results and Discussions.**    Table 5 shows that increasing location noise (corresponding to smaller $\kappa$ values) consistently leads to lower model performance. This trend highlights that geo-alignment is not optional but essential: when the model cannot align RS and SV features to the correct spatial anchor, its predictive power deteriorates. The performance degradation under higher noise highlights the importance of precise geo-alignment: accurate coordinates enable consistent cross-modal fusion, whereas noisy locations disrupt this alignment. This experiment empirically supports our design choice of explicitly using geolocations to align different modalities in GAIR.

## A.7    BENCHMARK DETAILS

To comprehensively evaluate the performance of GAIR, we construct multiple benchmark tasks spanning street view imagery, remote sensing data, and geolocation-based predictions. The details of each benchmark are as below.

### A.7.1    STREET VIEW IMAGERY TASKS

These tasks use ground-level imagery to predict socio-economic indicators, human perception metrics, and image metadata.

- **Socio-Economic Indicator Regression:** This task is predicting ten socio-economic indicators using 410,286 street view images from Los Angeles and Boston. The indicators include population density, educational attainment (percentage of people with a bachelor's degree or higher), health condition rates, racial demographics (percentage of people of color), median household income, proximity to public transportation, percentage of people who walk or bike, proportion of the population over 65 years old, crime rates, and visible sky area.

- **Human Perception Regression:** Using labels from the Global Streetscapes dataset (Hou et al., 2024), this task predicts human perception ratings for six attributes—'Beautiful', 'Boring', 'Depressing', 'Lively', 'Safe', and 'Wealthy'. These scores range from 0 to 1 and capture subjective assessments of urban environments.

- **View Direction Classification:** This classification task, based on the Global Streetscapes dataset, involves predicting the viewing direction of a street view image. The dataset provides two labels: "Front/Back" and "Side."

- **Imaging Platform Classification:** This task classifies the type of imaging platform used to capture a street view image, using labels from the Global Streetscapes dataset. There 6 classes including driving surface, walking surface, cycling surface, tunnel, open fields, and railway.

### A.7.2    REMOTE SENSING TASKS.

These benchmarks evaluate the performance of models on various remote sensing-based tasks.

- **Burn Scars Estimation:** This task uses the HLS Burn Scars dataset (Jakubik et al., 2023), which contains Sentinel-2 imagery of burn scars with burn scars masks for the years 2018-2021 in the United States. The dataset contains 804 scenes, each of size $512 \times 512$ pixels with six spectral bands (Blue, Green, Red, NIR, SW1, SW2). The masks contain a single band, where pixels labeled 1 represent burn scars, and 0 indicate unaffected areas. The dataset is randomly split into 2/3 for training and 1/3 for validation.

- **Crop Type Mapping (South Sudan):** This task uses a crop-type semantic segmentation dataset from South Sudan (M Rustowicz et al., 2019), based on Sentinel-2 imagery from 2017. The dataset covers 837 agricultural fields and includes four crop types. Each Sentinel-2 image is of $64 \times 64$ pixels and consists of 10 spectral bands (B2-B8, B8A, B11, and B12).

- **Cropland Boundary Delineation:** We employ the AI4SmallFarms dataset (Persello et al., 2023) for cropland boundary detection. This dataset includes 439,001 manually annotated agricultural field polygons distributed across 62 non-overlapping tiles. Each tile is derived from Sentinel-2 imagery, utilizing four spectral bands (B2, B3, B4, and B8) for delineation.

### A.7.3 LOCATION-BASED TASKS

These benchmarks incorporate both geolocation information and remote sensing imagery to evaluate geospatial representation learning:

- **Geo-Aware Image Regression:** For image regression tasks, we mainly use the MOSAIKS dataset (Rolf et al., 2021; Yeh et al., 2021), which provides large-scale geospatial observations. The specific tasks include:

    - **Population Density:** This task estimates population density from daytime satellite imagery using the MOSAIKS dataset. The dataset originally consists of 100,000 records, but after preprocessing, we retain 425,637 geographically distributed samples. A log transformation is applied to normalize zero-valued cases.
    - **Forest Cover:** Leveraging remote sensing data, this task predicts the percentage of land covered by forests, defined as vegetation exceeding 5 meters in height. The dataset includes 498,106 observations globally.
    - **Nightlight Luminosity:** This task utilizes nighttime satellite images to predict the average radiance at night, as measured by the Visible Infrared Imaging Radiometer Suite (VIIRS) in 2015. The dataset contains 492,226 observations.
    - **Elevation:** Using remote sensing RGB bands, this task predicts elevation levels based on data from the Shuttle Radar Topography Mission (SRTM) at NASA's Jet Propulsion Laboratory (JPL) and other sources. The MOSAIKS dataset provides 498,115 elevation records.

- **Geo-Aware Species Recognition:** Geo-Aware fine-grained species recognition aims to classify images of different species into different fine-grained categories using both images and their location metadata. We include four datasets for evaluation:

    - **BirdSnap:** A dataset focused on bird species in North America, BirdSnap (Berg et al., 2014) contains 19,567 images across 500 species, with geolocation annotations provided by Mac Aodha et al. (2019).
    - **NABirds:** NABirds (Van Horn et al., 2015) includes 23,699 images spanning 555 bird species from North America. The location information is derived from the eBird dataset (Sullivan et al., 2009), which aggregates citizen-science bird observations.
    - **iNat2017:** This dataset, sourced from the iNaturalist 2017 challenge (Van Horn et al., 2018), contains 675,170 images across 5,089 species categories. Each image is associated with a location, covering global regions.
    - **iNat2018:** Similar to iNat2017, this dataset is part of the iNaturalist 2018 challenge (Van Horn et al., 2018), featuring 461,939 images across 8,142 species categories. It has a similar global distribution and data structure.

A.8 IMPLEMENTATION DETAILS

**Model Pretraining.** We train our model on a Linux server equipped with 4 NVIDIA RTX A6000 GPUs (48GB). The training is conducted with a batch size of 256, a base learning rate of $1.5 \times 10^{-6}$, and a warm-up of the first 5% of total epochs. We apply two types of data augmentations: (1) standard augmentations for both Sentinel-2 imagery and street view images, including random horizontal flipping and color jittering, and (2) temporal augmentation for Sentinel-2 imagery, where images from different timestamps are randomly selected as Ayush et al. (2021) did. We fix the input size of remote sensing image to be $96 \times 96$ and street view image to be $224 \times 224$; handling dynamic input sizes is left for future work. The loss balancing parameter $\lambda$ is set to 0.5. We use the AdamW optimizer with $\beta_1 = 0.9$, $\beta_2 = 0.999$, and a weight decay of 0.01. All models, including baselines and GAIR, adopt the ViT-B backbone; for the remote sensing encoder specifically, we use a smaller patch size of $8 \times 8$ to better capture fine-grained textures.

**Model Fine-Tuning.** In this work, we evaluate GAIR across three benchmarks: street view imagery benchmarks, remote sensing imagery benchmarks, and location benchmarks.

For street view imagery benchmarks, we apply three fine-tuning strategies to adapt the pretrained model to downstream tasks: 1) *Linear probing*: The encoder remains frozen, and only a single-layer MLP head is fine-tuned. 2) *Non-linear probing*: The encoder remains frozen, while the head consists of two MLP layers with a sigmoid activation function. 3) *Full fine-tuning*: All model parameters, including the encoder, are fine-tuned. For the remote sensing imagery benchmark, we evaluate the representation quality of the pretrained models by evaluating them with frozen encoders without further fine-tuning. For location benchmarks, given the smaller capacity of the location encoder, we use the pretrained model as parameter initialization and then perform full fine-tuning to evaluate its performance. All baselines follow the same experimental setup, the batch size is of 256, 50 training epochs, a learning rate of $1 \times 10^{-3}$, and all images are resized to match the input requirements of each baseline model.

Due to the lack of official model implementations for the street view image benchmark, we reimplement all the street view image baselines. For the remote sensing benchmark, we directly use remote sensing baselines from PANGAEA-Bench (Marsocci et al., 2024). Similarly, we use the location encoding baselines from TorchSpatial (Wu et al., 2024).

A.9 COMPUTATION COST

In this section, we will analyze the computation cost of the proposed NILI module in detail.

**Where the NILI cost appears.** The Neural Implicit Local Interpolation (NILI) module is only used during *pretraining* to learn geo-aligned representations. For downstream tasks, we discard the module and only keep the pretrained encoders $f(\cdot)$ (RS), $g(\cdot)$ (SV), and $e(\cdot)$ (Loc), incurring *no* NILI overhead at inference time. If one chooses optional joint multi-modal inference (Appendix A.13), NILI is re-enabled and introduces a small latency increase (Table 6).

**Overhead versus a pooling baseline.** Relative to a baseline that uses simple average pooling on the RS encoder feature map, NILI adds only **0.02 GFLOPs** during pretraining. In our setup this overhead is negligible compared to the ViT backbones and data I/O.

Table 6: Compute comparison for the core encoders with and without NILI. Inference time is measured per sample (batch-normalized) on a single A6000. Note that NILI is removed in all downstream tasks by default; the "multi-modal" line reports the optional joint inference case.

| Model | GFLOPs | Inference mode | Latency (ms) |
|---|---|---|---|
| GAIR *w/o NILI* | 23.10 | Encoders only (default) | 10.34 |
| GAIR *with NILI* | 23.12 | *Joint multi-modal (optional)* | 16.01 |

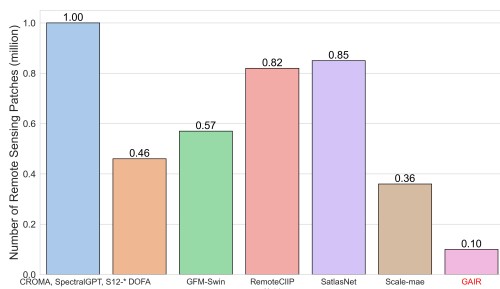

Figure 8: Comparison of the number of remote sensing patches used during model pretraining.

**End-to-end training cost.** Table 7 summarizes the per-epoch wall clock and the marginal overhead introduced by NILI during pretraining. As NILI runs only for coordinate-aligned contrastive pairs and uses constant-size neighborhoods, its added time remains a small fraction of the epoch time.

Table 7: Pretraining epoch time on 2×A6000 (batch size 256).

| Configuration | Epoch time (h) | Extra GFLOPs vs. GAIR *w/o NILI* |
|---|---|---|
| GAIR *w/o NILI* | ≈ 3.5 | – |
| GAIR *with NILI* | ≈ 3.5 | + 0.02 |

### A.10 THE COMPARISONS OF REMOTE SENSING TRAINING SAMPLES IN DIFFERENT GEOFMS

Figure 8 presents a comparative analysis of the number of remote sensing patches utilized during pretraining across various GeoFMs. Notably, GAIR is pretrained with only 0.1 million samples, significantly fewer than other models. Despite this reduced dataset size, GAIR achieves competitive performance, demonstrating the effectiveness of its joint training paradigm that integrates street view imagery with remote sensing data.

### A.11 RECONSTRUCTION LOSS AND CONTRASTIVE LOSS

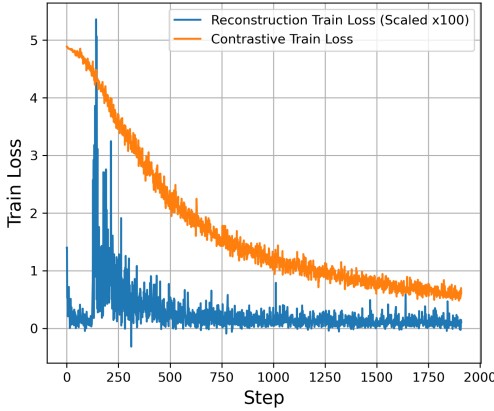

Figure 9: The loss curve of GAIR-MAE and GAIR during model pretraining.

Existing GeoFMs primarily adopt two pretraining objectives: reconstruction loss (Cong et al., 2022; Hong et al., 2024; Fuller et al., 2024; Xiong et al., 2024; Astruc et al., 2024) and contrastive loss (Hong et al., 2024; Jain et al., 2024). To investigate the effectiveness of these approaches, we introduce an ablation setting where the model learns to reconstruct missing image patches by treating the street view image as a masked region of the remote sensing image (Figure 7 in Appendix A.5). However, we observe that during pretraining, the loss remains unstable (Figure 9). ultimately leading the model to perform worse in downstream tasks.

A.12 COMPREHENSIVE EVALUATIONS ON THE STREET VIEW BENCHMARK

To provide a more comprehensive analysis of model performance beyond the overall metrics reported in the main paper (Table 1), we presents a detailed, per-indicator evaluation on the street view benchmark, as shown in Tables 8 to 13 which include experimental results for 6 human perception indicators and 10 socio-economic indicators.

Table 8: Detailed non-linear probing results on human perception prediction (RMSE ↓). **Bold** indicates the best result.

| Model | Wealthy | Safe | Beautiful | Depressing | Boring | Lively | Average |
|---|---|---|---|---|---|---|---|
| SatMAE | 2.1279 | 2.0132 | 2.2863 | 1.9920 | 2.0173 | 2.2809 | 2.1196 |
| CROMA | 2.0435 | 1.9309 | 2.1340 | 1.9417 | 1.9464 | 2.1350 | 2.0219 |
| PIS | 1.8885 | 1.8291 | 1.9605 | 1.8354 | 1.8693 | 1.9362 | 1.8865 |
| Random Init. | 2.1763 | 2.0402 | 2.3127 | 2.0057 | 2.0397 | 2.3458 | 2.1534 |
| ImageNet Init. | **1.6020** | 1.5974 | 1.7198 | 1.6543 | 1.6659 | **1.6240** | **1.6439** |
| MoCo v3-ImageNet | 1.6168 | 1.6058 | 1.7301 | 1.6868 | 1.6893 | 1.6306 | 1.6599 |
| MAE-ImageNet | 1.7267 | 1.7162 | 1.8373 | 1.7507 | 1.7696 | 1.7547 | 1.7592 |
| MoCo v3-Streetscapes | 1.7234 | 1.6883 | 1.7289 | 1.6943 | **1.6595** | 1.7560 | 1.7084 |
| MAE-Streetscapes | 1.7489 | 1.7014 | 1.8499 | 1.7891 | 1.7933 | 1.7379 | 1.7701 |
| GAIR-MAE | 2.1034 | 1.9379 | 2.2049 | 2.0911 | 2.0021 | 1.9576 | 2.0495 |
| GAIR w/o Loc | 1.6299 | 1.6252 | 1.7392 | 1.6687 | 1.6673 | 1.7389 | 1.6782 |
| GAIR | 1.6046 | **1.5837** | **1.7044** | **1.6533** | 1.6609 | 1.6865 | 1.6489 |

Table 9: Detailed fully fine-tune results on human perception prediction (RMSE ↓). **Bold** indicates the best result.

| Model | Wealthy | Safe | Beautiful | Depressing | Boring | Lively | Average |
|---|---|---|---|---|---|---|---|
| SatMAE | 1.8675 | 1.8129 | 1.9498 | 1.9920 | 1.8515 | 1.8134 | 1.8812 |
| CROMA | 1.8911 | 1.8487 | 1.9577 | 1.9417 | 1.8650 | 2.1280 | 1.9387 |
| PIS | 1.5791 | 1.5985 | 1.6760 | 1.8354 | 1.6270 | 1.3835 | 1.6166 |
| Random Init. | 2.1840 | 2.0950 | 2.3662 | 2.0057 | 2.1149 | 2.3376 | 2.1839 |
| ImageNet Init. | 1.4851 | **1.5008** | 1.5831 | 1.6543 | 1.5466 | 1.3929 | 1.5271 |
| MoCo v3-ImageNet | 1.5503 | 1.5532 | 1.6372 | 1.6868 | 1.5913 | 1.4938 | 1.5821 |
| MAE-ImageNet | 1.5508 | 1.5442 | 1.6407 | 1.7507 | 1.5897 | 1.3966 | 1.5788 |
| MoCo v3-Streetscapes | 1.5683 | 1.5453 | 1.6292 | 1.6943 | 1.5882 | 1.4547 | 1.5800 |
| MAE-Streetscapes | 1.5524 | 1.5299 | 1.6288 | 1.7891 | 1.5813 | 1.4830 | 1.5774 |
| GAIR-MAE | 1.9277 | 1.8892 | 1.9837 | 1.9758 | 1.9023 | 2.0879 | 1.9611 |
| GAIR w/o Loc | 1.5033 | 1.5473 | 1.6044 | 1.6613 | 1.6144 | 1.6453 | 1.5960 |
| GAIR | **1.4790** | 1.5021 | **1.5582** | **1.6321** | **1.5253** | **1.3465** | **1.5072** |

Table 10: Detailed linear probing results on human perception prediction (RMSE ↓). **Bold** indicates the best result.

| Model | Wealthy | Safe | Beautiful | Depressing | Boring | Lively | Average |
|---|---|---|---|---|---|---|---|
| SatMAE | 2.1928 | 2.0501 | 2.3379 | 2.0167 | 2.0441 | 2.3669 | 2.1681 |
| CROMA | 2.1004 | 1.9736 | 2.2198 | 1.9784 | 1.9861 | 2.2026 | 2.0768 |
| PIS | 2.0057 | 1.9110 | 2.0706 | 1.9257 | 1.9353 | 2.0665 | 1.9858 |
| Random Init. | 2.2451 | 2.0706 | 2.3478 | 2.0294 | 2.0647 | 2.4424 | 2.2000 |
| ImageNet Init. | 1.7000 | 1.7063 | 1.8411 | 1.7430 | 1.7598 | 1.7337 | 1.7473 |
| MoCo v3-ImageNet | 1.7229 | 1.7181 | 1.8588 | 1.7701 | 1.7583 | 1.7259 | 1.7590 |
| MAE-ImageNet | 1.8577 | 1.8164 | 1.9840 | 1.8410 | 1.8608 | 1.8781 | 1.8730 |
| MoCo v3-Streetscapes | 1.7395 | 1.7325 | 1.8688 | 1.7801 | 1.7666 | 1.6731 | 1.7601 |
| MAE-Streetscapes | 1.8489 | 1.7901 | 1.9501 | 1.8633 | 1.8418 | 1.8142 | 1.8514 |
| GAIR-MAE | 2.0091 | 2.4948 | 2.2232 | 2.0929 | 2.0375 | 1.8949 | 2.1254 |
| GAIR w/o Loc | **1.6822** | 1.6932 | 1.8342 | 1.7483 | **1.7400** | 1.7859 | 1.7473 |
| GAIR | 1.6854 | **1.6901** | **1.8201** | **1.7299** | 1.7428 | **1.6163** | **1.7141** |

## A.13 Ablation Studies on the Benefits of Multimodal Fusion

In addition, we investigate the impact of different modality combinations on the model prediction accuracy for these 10 socio-economic indicators. More specifically, we compare three single-modality models by using the street view image encoder alone, denoted as GAIR-MAE, GAIRw/o Loc, and GAIR, and the single-modality model by using the RS image encoder alone, denoted as GAIR- (RS Only), against multimodal fusion approaches including GAIR (RS+SV), GAIR (SV+Loc), and GAIR (SV+Loc+RS). GAIR (RS+SV) indicates that we use GAIR's RS image encoder and streetview image encoder to extract geographically colocated RS image embedding and SV image embedding at location $x_i$. These two image embeddings are concatenated and fed into a probing head for socio-economic indicator prediction. Similarly, GAIR (SV+Loc) indicates using the SV image encoder and location encoder to perform the prediction, and GAIR (SV+Loc+RS) uses all three encoders for these prediction tasks.

Due to computational constraints, this modality analysis focuses on the socio-economic tasks. The experimental results are shown in Tables 11, 12, and 13. The results consistently demonstrate the benefit of integrating multiple data sources. Specifically, the full multimodal fusion, i.e., GAIR (SV+Loc+RS), achieves the lowest average RMSE across all three evaluation settings: non-linear probing (0.6611), full fine-tuning (0.6012), and linear probing (0.6678). This represents a significant error reduction compared to single or dual modality approaches. For instance, compared with GAIR (RS only), and SV only baseline denoted as GAIR, the full fusion reduces the average RMSE by approximately 20% across all settings. Even when compared against the strongest dual-modality baseline in each setting – GAIR (SV+Loc), the full SV+Loc+RS fusion approach, GAIR (SV+Loc+RS), provides a further average RMSE reduction ranging from 4% to 15.1%, indicating the benefits of multimodal fusion

Table 11: Detailed non-linear probing results on socio-economic indicator prediction (RMSE ↓). Multi-model results are also included here. **Bold** indicates the best result. "Health Con." indicates health condition rate. "Pop. Den." indicates population density. "Edu. Att." indicates educational attainment. "Racial" indicates the racial demographics. "Med. Income" indicates the median household income. "Pub. Tra." indicates the proximity to public transportation. "Pop. > 65" indicates Population over 65. "% walk/bike" indicates the percentage of walking or biking.

| Model | Health Con. | Pop. Den. | Edu. Att. | Racial | Med. Income | Pub. Tra. | Crime Rate | Sky Area | Pop. > 65 | % walk /bike | Average |
|---|---|---|---|---|---|---|---|---|---|---|---|
| SatMAE | 1.0076 | 0.9503 | 0.8924 | 0.9611 | 0.9263 | 0.9782 | 0.9699 | 1.0727 | 0.9621 | 0.9504 | 0.9671 |
| CROMA | 1.0060 | 0.9232 | 0.8829 | 0.9471 | 0.9118 | 0.9588 | 0.9573 | 1.0741 | 0.9546 | 0.9341 | 0.9550 |
| PIS | 0.9982 | 0.8591 | 0.8429 | 0.9112 | 0.8712 | 0.9087 | 0.9129 | 1.1550 | 0.9328 | 0.8979 | 0.9290 |
| Random Init. | 1.0082 | 0.9637 | 0.9109 | 0.9691 | 0.9395 | 0.9901 | 0.9753 | 1.0225 | 0.9671 | 0.9537 | 0.9700 |
| ImageNet Init. | 0.9675 | 0.7970 | 0.7639 | 0.8500 | 0.8038 | 0.8317 | 0.8251 | 1.3712 | 0.8965 | 0.8224 | 0.8929 |
| MoCo v3-ImageNet | 0.9737 | 0.7906 | 0.7523 | 0.8414 | 0.7972 | 0.8282 | 0.8242 | 1.2318 | 0.8956 | 0.8249 | 0.8760 |
| MAE-ImageNet | 0.9783 | 0.8021 | 0.7723 | 0.8587 | 0.8123 | 0.8441 | 0.8398 | 1.1624 | 0.9058 | 0.8403 | 0.8816 |
| MoCo v3-Streetscapes | 0.9746 | 0.8011 | 0.7684 | 0.8499 | 0.8030 | 0.8293 | 0.8346 | 1.2715 | 0.8994 | 0.8312 | 0.8863 |
| MAE-Streetscapes | 0.9898 | 0.7967 | 0.7747 | 0.8694 | 0.8011 | 0.8341 | 0.8292 | 1.3039 | 0.9018 | 0.8393 | 0.8940 |
| GAIR-MAE | 1.0050 | 0.9602 | 0.9011 | 0.9713 | 0.9270 | 0.9895 | 0.9502 | 0.8840 | 0.9456 | 0.9510 | 0.9485 |
| GAIR w/o Loc | 0.9788 | 0.8013 | 0.7534 | 0.8552 | 0.7971 | 0.8213 | 0.8523 | 1.2329 | 0.9012 | 0.8301 | 0.8823 |
| GAIR | 0.9613 | 0.7981 | 0.7412 | 0.8688 | 0.8053 | 0.8246 | 0.8255 | 1.2423 | 0.9017 | 0.8342 | 0.8803 |
| GAIR (RS only) | 0.9845 | 0.8526 | 0.8160 | 0.8917 | 0.8456 | 0.8805 | 0.8573 | 0.7865 | 0.9202 | 0.8512 | 0.8686 |
| GAIR (RS+SV) | 0.9827 | 0.7998 | 0.7646 | 0.8637 | 0.8000 | 0.8552 | 0.8293 | 0.7158 | 0.9051 | 0.8298 | 0.8346 |
| GAIR (SV+Loc) | 0.8978 | 0.7266 | 0.5031 | 0.6287 | 0.6518 | 0.3934 | 0.7319 | 1.7467 | 0.8358 | 0.6742 | 0.7790 |
| GAIR (SV+Loc+RS) | **0.8857** | **0.7077** | **0.4933** | **0.6215** | **0.6384** | **0.3916** | **0.7023** | **0.6815** | **0.8284** | **0.6607** | **0.6611** |

## A.14 Geographic Bias Quantification and Debiasing of GAIR

Our Streetscapes1M dataset consists of tuples containing street view images, their geolocations, and corresponding RS images. Due to the data collection process of street view imagery, its geographic coverage is much denser in urban areas and considerably sparser in rural regions. This imbalance results in a strong urban–rural skew within Streetscapes1M. Consequently, pretraining GAIR exclusively on this dataset risks amplifying geographic bias Wu et al. (2024); Manvi et al. (2024). Thus, to debias GAIR across the geographic space, we collect additional 200,000 training samples from remote sensing imagery and corresponding location metadata to form a complementary pretraining dataset with urban-rural balanced RS-location pairs (as shown in Figure 10). We further

Table 12: Detailed fully fine-tune results on socio-economic indicator prediction (RMSE ↓). Multi-model results are also included here. **Bold** indicates the best result. The column names indicate the same meaning as those in Table 11.

| Model | Health Con. | Pop. Den. | Edu. Att. | Racial | Med. Income | Pub. Tra. | Crime Rate | Sky Area | Pop. > 65 | % walk /bike | Average |
|---|---|---|---|---|---|---|---|---|---|---|---|
| SatMAE | 0.8968 | 0.7684 | 0.6929 | 0.7779 | 0.7488 | 0.7419 | 0.7440 | 0.7042 | 0.8504 | 0.7647 | 0.7690 |
| CROMA | 0.9174 | 0.7849 | 0.7185 | 0.8046 | 0.8650 | 0.7723 | 0.7907 | 0.6230 | 0.8650 | 0.7857 | 0.7927 |
| PIS | 0.8001 | 0.6895 | 0.5940 | 0.6915 | 0.7922 | 0.6391 | 0.6445 | 0.5034 | 0.7922 | 0.6505 | 0.6797 |
| Random Init. | 0.9991 | 0.8906 | 0.8499 | 0.9206 | 0.8840 | 0.9262 | 0.9320 | 0.7820 | 0.9413 | 0.9093 | 0.9035 |
| ImageNet Init. | 0.8207 | 0.7056 | 0.6167 | 0.7056 | 0.6854 | 0.6573 | 0.6831 | 0.6583 | 0.8001 | 0.6793 | 0.7012 |
| MoCo v3-ImageNet | 0.7846 | **0.6886** | 0.5957 | 0.6886 | 0.6687 | 0.6290 | 0.6552 | 0.6358 | 0.7831 | 0.6497 | 0.6779 |
| MAE-ImageNet | 0.8335 | 0.7056 | 0.6356 | 0.7203 | 0.7027 | 0.6798 | 0.6942 | 0.6711 | 0.8145 | 0.6927 | 0.7150 |
| MoCo v3-Streetscapes | 0.7934 | 0.7035 | 0.6093 | 0.6903 | 0.7044 | 0.6895 | 0.6395 | **0.4288** | 0.7889 | 0.6593 | 0.6707 |
| MAE-Streetscapes | 0.8042 | 0.7185 | 0.6286 | 0.7094 | 0.7293 | 0.6663 | 0.6499 | 0.5243 | 0.7813 | 0.6601 | 0.6872 |
| GAIR-MAE | 0.9001 | 0.7785 | 0.7081 | 0.7832 | 0.7934 | 0.7933 | 0.8088 | 0.8038 | 0.8694 | 0.8092 | 0.8048 |
| GAIR w/o Loc | 0.7732 | 0.6945 | 0.6204 | 0.7012 | 0.6684 | 0.6103 | **0.6234** | 0.5974 | 0.7763 | 0.6597 | 0.6725 |
| GAIR | 0.7843 | 0.7051 | 0.5982 | 0.6783 | 0.6589 | 0.6295 | 0.6300 | 0.4960 | 0.7894 | 0.6422 | 0.6612 |
| GAIR (RS only) | 0.8666 | 0.7595 | 0.6594 | 0.7305 | 0.7468 | 0.6331 | 0.7790 | 0.7293 | 0.8342 | 0.6785 | 0.7417 |
| GAIR (RS+SV) | 0.7653 | 0.7578 | 0.6531 | 0.7245 | 0.7394 | 0.6325 | 0.7774 | 0.6271 | 0.7340 | 0.6079 | 0.7019 |
| GAIR (SV+Loc) | **0.7216** | 0.7501 | 0.5175 | 0.6396 | 0.6646 | 0.5845 | 0.7830 | 0.8608 | 0.5110 | **0.5063** | 0.6539 |
| GAIR (SV+Loc+RS) | 0.7453 | 0.7023 | **0.5022** | **0.5422** | **0.6012** | **0.4577** | 0.6922 | 0.7383 | **0.5012** | 0.5293 | **0.6012** |

Table 13: Detailed linear probing results on socio-economic indicator prediction (RMSE ↓). Multi-model results are also included here. **Bold** indicates the best result. The column names indicate the same meaning as those in Table 11.

| Model | Health Con. | Pop. Den. | Edu. Att. | Racial | Med. Income | Pub. Tra. | Crime Rate | Sky Area | Pop. > 65 | % walk /bike | Average |
|---|---|---|---|---|---|---|---|---|---|---|---|
| SatMAE | 1.0079 | 0.9674 | 0.9270 | 0.9745 | 0.9501 | 0.9932 | 0.9787 | 0.9308 | 0.9725 | 0.9690 | 0.9671 |
| CROMA | 1.0067 | 0.9468 | 0.9155 | 0.9645 | 0.9363 | 0.9770 | 0.9696 | 0.9111 | 0.9652 | 0.9574 | 0.9550 |
| PIS | 1.0025 | 0.8916 | 0.8832 | 0.9377 | 0.9012 | 0.9339 | 0.9456 | 0.9191 | 0.9450 | 0.9303 | 0.9290 |
| Random Init. | 1.0083 | 0.9718 | 0.9359 | 0.9784 | 0.9538 | 0.9981 | 0.9816 | 0.9273 | 0.9738 | 0.9710 | 0.9700 |
| ImageNet Init. | 0.9927 | 0.8520 | 0.8457 | 0.9150 | 0.8673 | 0.8968 | 0.9145 | 0.8100 | 0.9321 | 0.9031 | 0.8929 |
| MoCo v3-ImageNet | 0.9916 | 0.8344 | 0.8148 | 0.8957 | 0.8454 | 0.8849 | 0.9083 | 0.7760 | 0.9217 | 0.8873 | 0.8760 |
| MAE-ImageNet | 0.9928 | 0.8432 | 0.8251 | 0.9034 | 0.8578 | 0.8867 | 0.9030 | 0.7882 | 0.9267 | 0.8892 | 0.8816 |
| MoCo v3-Streetscapes | 0.9888 | 0.8235 | 0.8283 | 0.8955 | 0.8479 | 0.8685 | 0.8982 | 0.8928 | 0.9399 | 0.8794 | 0.8863 |
| MAE-Streetscapes | 1.0004 | 0.8185 | 0.8385 | 0.9102 | 0.8422 | 0.8677 | 0.8900 | 0.9696 | 0.9288 | 0.8740 | 0.8940 |
| GAIR-MAE | 0.9996 | 0.9574 | 0.9267 | 0.9725 | 0.9807 | 0.9767 | 0.9422 | 0.8292 | 0.9699 | 0.9300 | 0.9485 |
| GAIR w/o Loc | 0.9783 | 0.8623 | 0.8300 | 0.8868 | 0.8357 | 0.8881 | 0.8872 | 0.8528 | 0.9252 | 0.8766 | 0.8823 |
| GAIR | 0.9844 | 0.8254 | 0.8585 | 0.8863 | 0.8289 | 0.8883 | 0.8810 | 0.8513 | 0.9278 | 0.8710 | 0.8803 |
| GAIR (RS only) | 0.9928 | 0.8742 | 0.8422 | 0.9120 | 0.8638 | 0.9008 | 0.9127 | 0.9056 | 0.9275 | 0.8862 | 0.8918 |
| GAIR (RS+SV) | 0.9877 | 0.8228 | 0.7889 | 0.8848 | 0.8227 | 0.8720 | 0.8940 | 0.7292 | 0.9101 | 0.8609 | 0.8573 |
| GAIR (SV+Loc) | 0.9299 | 0.7460 | 0.5245 | 0.8481 | 0.6763 | 0.8085 | 0.7849 | 0.0997 | 0.8481 | 0.7094 | 0.6976 |
| GAIR (SV+Loc+RS) | **0.9241** | **0.7310** | **0.5149** | **0.6425** | **0.6617** | 0.8051 | **0.7700** | **0.0873** | **0.8408** | **0.7009** | **0.6678** |

pretrain the remote sensing encoder and location encoder of GAIR on this dataset for 10 epochs while excluding the SV encoder due to the unavailability of SV images in rural areas. The resulting model, namely **GAIR *Debias***, is compared with the original GAIR, and other baselines (e.g., GeoCLIP (Vivanco Cepeda et al., 2024) and Taxabind (Sastry et al., 2025)) on multiple location benchmark datasets used in Table 3. We evaluate the overall model performance of these models as well as their geographic bias with the established Geo-Bias score, namely the *marked SSI score*, proposed by Wu et al. (2024).

While the overall accuracy on the location benchmark remains largely unchanged (see Table 3), we observe a clear reduction in geographic bias, measured by the Geo-Bias Score (Wu et al., 2024) as shown in Figure 4. Here, lower geo-bias, i.e., lower marked SSI scores, indicates less disparity of model performance across different places. This demonstrates that leveraging broader RS and location data can mitigate

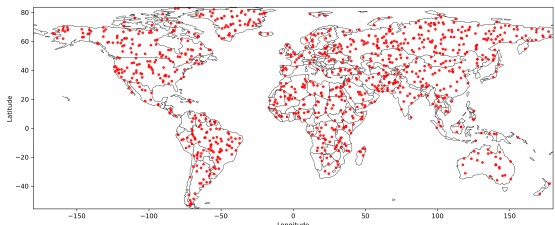

Figure 10: The distribution of additional training samples. We randomly generate 1,000 global locations, and for each location collect 200 remote sensing patches with their coordinates to form the training dataset.

Table 14: Zero-shot geolocalization accuracy of GAIR on 15k sampled SV queries.

| Method | Acc@1km | Acc@25km | Acc@200km | Acc@750km | Acc@2500km |
|---|---|---|---|---|---|
| GeoCLIP(Vivanco Cepeda et al., 2024) | 9.45% | 18.48% | 30.11% | 38.99% | 45.17% |
| GAIR | 10.04% | 21.35% | 33.15% | 40.11% | 51.65% |

geographic bias even without the supervision signal of street view images.

Note that we decided to use these location benchmark datasets from the LocBench (Wu et al., 2024) for geographic bias evaluation instead of the remote sensing benchmark datasets shown in Table 2. The reason is that all these location benchmark datasets have global geographic coverage, making them suitable for geographic bias evaluation. In contrast, most datasets used for RS image semantic segmentation and instance segmentation only cover a limited geographic area and thus unsuitable for this purpose, including the HLS Burns dataset for burn scar segmentation (Jakubik et al., 2023), the cropland polygon delineation dataset (Persello et al., 2023), and the crop type mapping dataset (M Rustowicz et al., 2019) listed in Table 2.

## A.15 ZERO-SHOT GEOLOCALIZATION EXPERIMENT

**Task Definition.** We design a zero-shot image geolocalization evaluation to test whether the pretrained encoders of GAIR can recover the approximate location of a street view (SV) image without task-specific fine-tuning. Given a query SV image $s_i$, the goal is to predict its geographic coordinate $x_i$ by comparing the embedding of $s_i$ with a gallery of location embeddings.

**Evaluation Protocol.** We randomly sample 15,000 SV images from the Streetscapes1M dataset as queries. For each query, we:

1. Use the SV encoder $g(\cdot)$ to extract an image embedding $g(s_i)$.
2. Use the location encoder $e(\cdot)$ to encode the GPS coordinates of all gallery images into a location embedding set $\{e(x_j)\}$.
3. Compute cosine similarity between $g(s_i)$ and each $e(x_j)$.
4. Select the top-1 most similar location embedding as the predicted location $\hat{x}_i$.

We compute the great-circle distance $d(x_i, \hat{x}_i)$ between the ground-truth location $x_i$ and the predicted location $\hat{x}_i$ using the haversine formula. Accuracy is reported under multiple distance thresholds:

$$\text{Acc@}\, r = \frac{1}{N} \sum_{i=1}^{N} 1\left[d(x_i, \hat{x}_i) < r\right] \tag{5}$$

where $r \in \{1, 25, 200, 750, 2500\}$ km. This follows common practice in global image geolocalization models such as GeoCLIP (Vivanco Cepeda et al., 2024).

**Results and Discussions.** Table 14 summarizes results on the sampled subset. GAIR achieves higher accuracy than GeoCLIP across all distance thresholds. Performance is particularly strong at coarse spatial scales ($> 200$ km), while fine-grained localization at the 1 km level remains very strong. These results demonstrate that GAIR learns transferable, location-aware representations effective for retrieval-based geospatial tasks.

## A.16 MORE VISUALIZATION RESULTS OF SPATIAL ALIGNMENT

Figure 11 and 12 show more visualization results of spatial alignment.

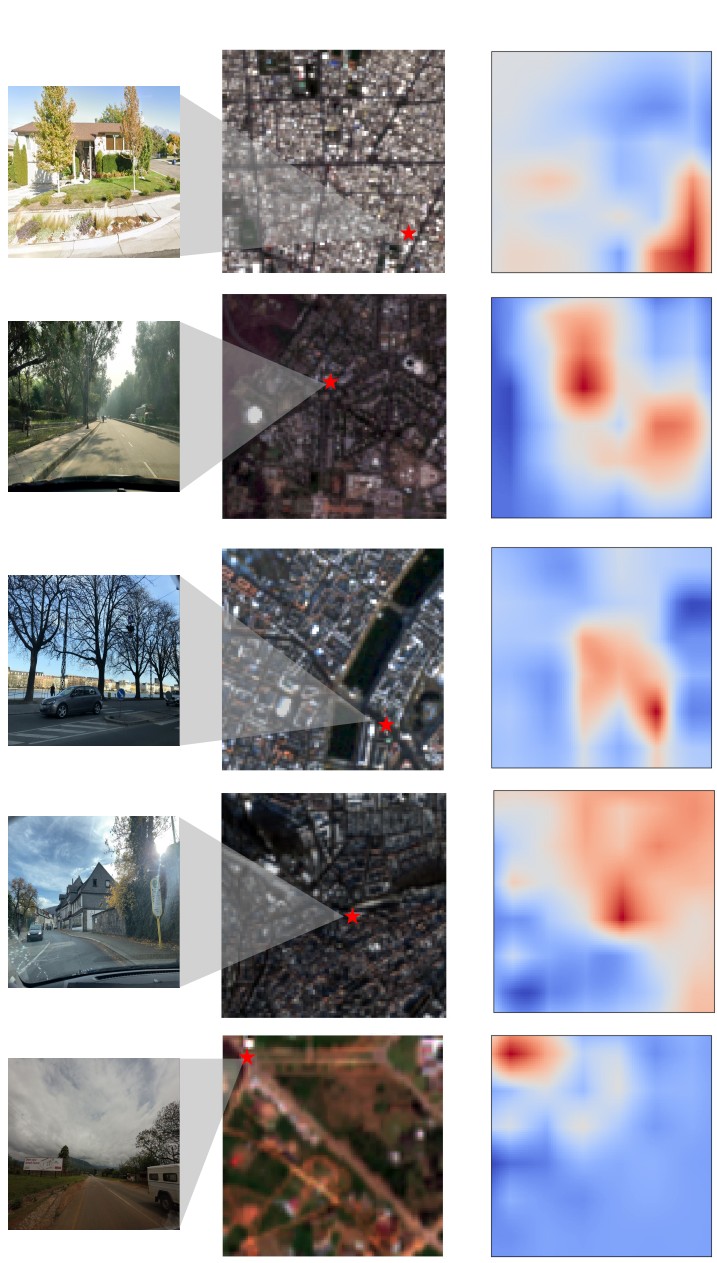

Figure 11: More results of cosine similarities between a SV image embedding $g(s_i)$ and different localized RS image embeddings $z_i^{(q)}$. The spatial scale of the similarity map is $100 \times 100$ pixels

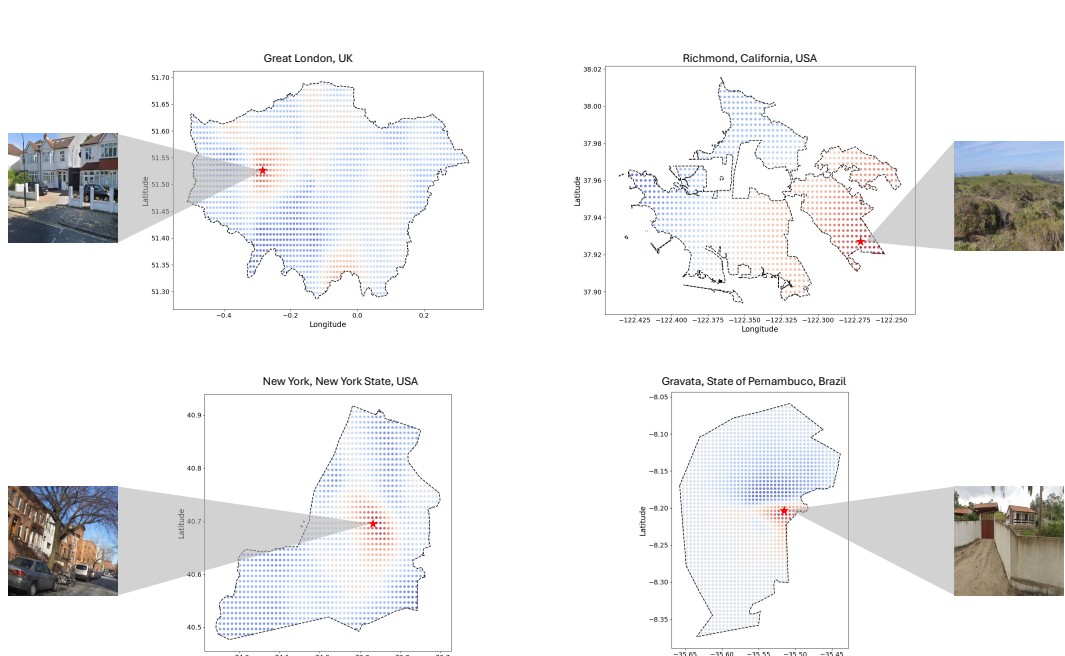

Figure 12: More results of cosine similarities between a SV image embedding $g(s_i)$ and location embeddings $e(x_i)$.

