# OpenReview forum: "GAIR: A Multimodal Geo-Foundation Model with Geo-Aligned Implicit Representations"
_ICLR.cc/2026/Conference — ICLR 2026 Conference Withdrawn Submission_

### Official Review · Reviewer_fMPm · 2025-10-25

**Soundness:** 3
**Presentation:** 3
**Contribution:** 3
**Rating:** 6
**Confidence:** 5

**Summary:**

SUMMARY: The authors propose GAIR, a new pretrained model consisting of contrastive pretraining of geographically aligned satellite (overhead) and street view (ground level) imagery. The model include a location encoder network that allows users at inference time to query location-specific embeddings capturing location characteristics only using latitude/longitude inputs. The model is trained using CLIP-style image-image and image-location matching. This work directly expands existing work on pretrained location encoding, namely SatCLIP and GeoCLIP, into a multi-modal direction. GAIR is evaluated extensively; all three encoders (satellite image, street view image and location) are evaluated and tested separately. The authors show improvements over existing models in all 3 categories.

**Strengths:**

STRENGTHS:
Some aspects I particularly enjoyed about this paper are:

- The paper is well motivated, both from a methodological and an application perspective. Working with geo-aligned satellite and on-the-ground imagery is interesting and being able to "translate" between these modalities / have embeddings based on different scales could be potentially very impactful.

- The experimental section of the paper is extensive! This is great to see; all 3 different encoders are evaluated. It might be a bit too expensive given that the authors sacrifice giving more details about the methodology.

- It's great that this work discusses and addresses data imbalances in geospatial data, esp. the urban-rural divide.

**Weaknesses:**

SHORTCOMINGS:

Major:

- The paper is heavily based on the use of implicit neural representations for geospatial applications, yet fails to cite foundational work in this space: [1] are the first to propose the use of INRs to learn continuous functions on the sphere of Earth while SatCLIP (which is cited) expands this approach to learning continuous Earth embeddings. This should be fixed!

- The choice of the location encoder architecture is not explored enough. The authors use a fourier-based location encoder architecture (as GeoCLIP does), however they don't go into a lot of detail on the exact design of the encoder. Furthermore, [1] also show that sin/cos transforms are not ideal for Earth data and a more specialized transform like Spherical Harmonics can perform better. It would be interesting to see different location encoder methods tested here.

- There is also a great lack of detail on the design choices for the satellite image and street view image encoder. How do these models look like exactly? The authors mention that they use a pretrained satellite image encoder - which one is that? It is hard to objectively assess the experimental performance without those details.

Minor:

- line 224: should be "multilayer perceptron"

- Performance gains are not huge and hard to assess without confidence intervals.

- Not enough discussion of limitations; conclusion section is super short.

**Questions:**

Overall this is an interesting paper with some interesting methodological advances and a great experimental section. I would ask the authors to address my concerns and questions outlined in the "Weaknesses" sections.

References:

[1] Rußwurm, Marc, et al. "Geographic location encoding with spherical harmonics and sinusoidal representation networks." arXiv preprint arXiv:2310.06743 (2023).

---

### Official Review · Reviewer_jEJB · 2025-10-31

**Soundness:** 2
**Presentation:** 2
**Contribution:** 2
**Rating:** 2
**Confidence:** 5

**Summary:**

The paper proposes GAIR (Geo-Aligned Implicit Representations), a multimodal Geo-Foundation Model (GeoFM) that integrates remote sensing (RS) imagery, street-view (SV) imagery, and geolocation metadata. GAIR extends ViT-based encoders with an implicit neural representation (INR) module called Neural Implicit Local Interpolation (NILI) to achieve continuous, fine-grained spatial alignment between overhead and ground-level modalities. The model is pretrained on the Streetscapes1M dataset using contrastive self-supervised learning and evaluated on 9 tasks across 22 datasets, including RS segmentation, SV regression/classification, and location-based prediction. The authors claim GAIR outperforms existing GeoFMs such as SatMAE, CROMA, DOFA, and TaxaBind in overall performance

**Strengths:**

Problem motivation: Addresses spatial alignment challenges in multimodal geospatial modeling by leveraging continuous representations rather than patch-level embeddings.

Model integration: Combines RS, SV, and geolocation embeddings in a unified self-supervised framework, potentially useful for large-scale geospatial representation learning.

Comprehensive experiments: Evaluations span diverse tasks (RS, SV, location) with ablation and bias analyses, which demonstrates methodological completeness.

Data contribution: The Streetscapes1M dataset may be useful to the broader community if released, as few large-scale RS–SV paired datasets exist.

**Weaknesses:**

Limited novelty and incremental contribution.
The method mainly combines existing components—ViT backbones, INR interpolation (essentially adapted from LIIF), and standard contrastive objectives—without introducing a fundamentally new mechanism. The “geo-alignment” through implicit interpolation is a direct application of known implicit coordinate mappings; no theoretical or algorithmic advancement beyond straightforward integration is shown.

Insufficient comparison and contextualization.
The paper barely situates GAIR relative to the rich literature of multimodal foundation models (e.g., OmniSat, GeoChat, RemoteCLIP, EarthGPT, SpectralGPT). Many relevant baselines are omitted or only superficially compared. No discussion is given on how GAIR differs architecturally or conceptually from these prior GeoFMs, especially those already using spatial alignment or contrastive geo-supervision.

Weak empirical evidence for innovation.
The performance gains over baselines are marginal and inconsistent across tasks (often within 0.5–1 mIoU or 0.05 F1), insufficient to justify a new foundation model. The “implicit alignment” advantage is not validated via spatial offset experiments or perturbation tests.

Presentation quality.
Many figures (e.g., Figures 1–3 and 5) are too small, with unreadable text, axes, and legends. Key architectural diagrams lack sufficient clarity to convey technical differences, making it difficult to follow model flow or understand parameter interactions.

Clarity and writing issues.
Important methodological descriptions (e.g., NILI equation details, coordinate projection Ψ, and training hyperparameters) are terse and sometimes ambiguous. Cross-references to appendices are excessive, and terminology such as “Geo-Foundation Model” and “implicit interpolation” is used without clear distinction from prior work.

**Questions:**

How does GAIR fundamentally differ from previous GeoFMs like OmniSat, DOFA, or GeoChat that already incorporate multimodal and geo-aligned embeddings?

Can the authors conduct spatial perturbation or registration-offset experiments to empirically verify that NILI indeed improves geo-alignment robustness?

The evaluation omits major multimodal FMs (e.g., RemoteCLIP, EarthGPT). Why were these not compared directly?

How sensitive is GAIR to coordinate noise and scale mismatch between RS and SV data?

Please provide enlarged, legible figures and detailed ablation visualizations to support interpretability.

---

### Official Review · Reviewer_ePnY · 2025-11-01

**Soundness:** 3
**Presentation:** 4
**Contribution:** 3
**Rating:** 6
**Confidence:** 5

**Summary:**

This paper proposes GAIR, a multimodal Geo-Foundation Model (GeoFM) that integrates overhead remote sensing (RS) imagery, street-view (SV) images, and geolocation metadata. The authors identify that Vision Transformers (ViTs) lack detailed localized image representations at arbitrary positions, which are essential for modeling geospatial relationships and aligning different modalities. To address this, they introduce an Implicit Neural Representation (INR) module called Neural Implicit Local Interpolation (NILI), which enables continuous spatial representations across the RS image. GAIR employs three factorized neural encoders and is trained with contrastive learning objectives on unlabeled multimodal data. The model is evaluated on 9 geospatial tasks and 22 datasets, demonstrating strong results compared with state-of-the-art GeoFMs and alternative objectives such as MoCo-v2 and MAE.

The contributions can be summarized as follows:
1. Defined the problem: the overhead RS image needs detailed localized image representations at arbitrary positions to align with other data modalities, like ground-level imagery and geospatial vector data (like longitude and latitude).
2. Proposed implicit neural representation (INR) module extending ViT with Neural Implicit Local Interpolation, which produces a continuous RS image representation covering arbitrary locations in the RS image
3. Proposed contrastive learning-based method to train a multimodal GeoFM GAIR to integrate overhead RS data, street view imagery, and geolocation metadata.
4. Evaluated across 9 geospatial tasks and 22 datasets (RS image-based, SV image-based, location embedding-based benchmarks), and achieved quite a shining result, and also included a lot of ablation studies to verify the effectiveness of their algorithms.

**Strengths:**

- The paper clearly defines the representation problem in multimodal geospatial learning and proposes a principled solution with the INR module.
- The NILI module is a creative idea that extends ViTs to continuous spatial domains.
- The experimental evaluation is extensive and well-structured, covering a wide range of tasks, modalities, and datasets.
- Ablation studies are comprehensive and provide evidence supporting the design choices.
- The writing is clear and the overall contribution is technically sound and reproducible.

**Weaknesses:**

- In the abstract, the claim that ViT “lacks detailed localized image representations at arbitrary positions” could be better justified: is this empirically validated or based on common understanding in the community?
- The term “generalizable geospatial representations” needs clarification — does it refer to zero-shot transfer or few-shot fine-tuning?
- Regarding the spatial interpolation in Figure 2(a) and Section 3.2, how can we verify that it actually works as intended? Interpolation from the four nearest patch embeddings assumes local linearity in representation space; please clarify if this assumption holds.
- Minor issue (L192–195): “path-level” should be “patch-level.”
- In the finetuning setup: “For remote sensing benchmarks, we fine-tune only the UPerNet while keeping the pretrained backbone frozen.” What motivates this choice? Is it due to the large parameter count of the RS encoder?
- For the street-view imagery benchmarks, why were CROMA, SatMAE, PIS, and TaxaBind chosen? Some of these (CROMA and SatMAE, for example, are trained on high resolution satellite imagery) are not pretrained on SV data, so the fairness of the comparison could be discussed.
- From the benchmark description, TaxaBind appears quite similar to GAIR; please further clarify the differences.
- The GAIR-MAE variant may introduce a viewpoint shift problem when treating SV images as masked RS patches. Is this pretraining setup feasible or well-justified?
- In Table 2, including U-Net and ViT baselines would improve completeness, as prior work (e.g., Pangaea benchmark) shows their competitiveness with GeoFMs.
- Figure 4 is unclear: what is “Masked SSI”? It is not defined in the paper.
- Similarly, Figure 5 does not clearly demonstrate the claimed alignment. The explanation of how GAIR learns geographical alignment should be expanded.

**Questions:**

Please refer to Weaknesses. Despite these questions and minor clarity issues, the submission is clear, technically correct, and experimentally rigorous. It presents a novel and well-motivated approach with strong empirical validation. I recommend acceptance, provided the authors address the raised clarifications.

---

### Official Review · Reviewer_pckc · 2025-11-01

**Soundness:** 2
**Presentation:** 2
**Contribution:** 2
**Rating:** 2
**Confidence:** 4

**Summary:**

The paper proposes a “Geo–foundation model”, which in practice consists of three separate components: an image encoder for satellite imagery, an image encoder for street-view imagery, and a location encoder mapping coordinates to embedding vectors. All three are trained jointly via CLIP-style objectives (Eqs. 2 and 3). Conceptually, the location encoder functions as an implicit neural representation, whereas the image encoders perform standard feature extraction.

However, the overall novelty is limited: the formulation essentially combines the geolocalization premise of GeoCLIP (street-view ↔ location) and SatCLIP (satellite image ↔ location).

Because the framework produces three different models, the experimental evaluation becomes complex and somewhat unfocused. Table 1 benchmarks the street-view component, Table 2 the satellite component, and Table 3 the location encoder. As such, the model does not seem to function as a single “foundation model”, but rather as a training framework for three separate ones.

**Strengths:**

* Unified framework to jointly train satellite, street-view, and coordinate encoders with a CLIP-style objective. I believe massive multi-modal pre-training beyond street-aereal-location should be explored more.
* Clear experimental setup demonstrating that each sub-model can be evaluated in isolation.

**Weaknesses:**

* Limited methodological novelty — largely a combination of ideas already present in SatCLIP and GeoCLIP.
* The “foundation model” framing is misleading: the method is effectively three separate models trained together, not a single unified model.
* Experimental evaluation feels scattered across sub-components with no strong, coherent take-home message.

**Questions:**

* How well motivated is the "implicit representation" aspect of this work? As I understand, only the the location encoder can be seens as an implicit neural representation, while image encoders perform regular feature extraction from image data.

---

### Note · Authors · 2025-12-27

I have read and agree with the venue's withdrawal policy on behalf of myself and my co-authors.